# HGNet: Scalable Foundation Model for Automated Knowledge Graph Generation from Scientific Literature

**Devvrat Joshi & Islem Rekik**
BASIRA Lab, Imperial-X (I-X) and Department of Computing
Imperial College London, London, United Kingdom
{devvrat.joshi24, i.rekik}@imperial.ac.uk

## Abstract

Automated knowledge graph (KG) construction is essential for navigating the rapidly expanding body of scientific literature. However, existing approaches face persistent challenges: they struggle to recognize long multi-word entities, often fail to generalize across domains, and typically overlook the hierarchical and logically constrained nature of scientific knowledge. While general-purpose large language models (LLMs) offer some adaptability, they are computationally expensive and yield inconsistent accuracy on specialized, domain-heavy tasks such as scientific knowledge graph construction. As a result, current KGs are shallow and inconsistent, limiting their utility for exploration and synthesis. We propose a two-stage framework for scalable, zero-shot scientific KG construction. The first stage, Z-NERD, introduces (i) Orthogonal Semantic Decomposition (OSD), which promotes domain-agnostic entity recognition by isolating semantic "turns" in text, and (ii) a Multi-Scale TCQK attention mechanism that captures coherent multi-word entities through n-gram–aware attention heads. The second stage, HGNet, performs relation extraction with hierarchy-aware message passing, explicitly modeling parent, child, and peer relations. To enforce global consistency, we introduce two complementary objectives: a Differentiable Hierarchy Loss to discourage cycles and shortcut edges, and a Continuum Abstraction Field (CAF) Loss that embeds abstraction levels along a learnable axis in Euclidean space. To the best of our knowledge, this is the first approach to formalize hierarchical abstraction as a continuous property within standard Euclidean embeddings, offering a simpler and more interpretable alternative to hyperbolic methods. To address data scarcity, we also release SPHERE[1], a large-scale, multi-domain benchmark for hierarchical relation extraction. Our framework establishes a new state of the art on benchmarks such as SciERC, SciER and SPHERE benchmarks, improving named entity recognition (NER) by 8.08% and relation extraction (RE) by 5.99% on the official out-of-distribution test sets. In zero-shot settings, the gains are even more pronounced, with improvements of 10.76% for NER and 26.2% for RE, marking a significant step toward reliable and scalable scientific knowledge graph construction.

## 1 Introduction

The exponential growth of scientific literature has created an overwhelming challenge: the pace of publication now far exceeds human capacity for manual review and synthesis Taylor et al. (2022). Automated systems that can distill unstructured text into structured, machine-readable representations are therefore essential. Knowledge Graphs (KGs) offer a compelling solution, representing entities such as methods, datasets, or concepts as nodes and their semantic connections as edges Wang et al. (2022a). Yet, constructing high-quality KGs from dense, jargon-rich scientific text re-

---

Code and data available at https://github.com/basiralab/HGNet
[1]Our benchmark dataset is available at https://github.com/basiralab/SPHERE

mains difficult, as complex terminology, long multi-word entities, and layered hierarchical structures introduce challenges that current approaches fail to resolve.

Scientific KG construction is constrained by four interdependent challenges that limit both accuracy and scalability. The first two concern node identification. Many scientific concepts are expressed as long multi-word phrases, such as *"in situ transmission electron microscopy"*, which must be recognized as coherent units. This problem of *multi-word entity recognition* remains unresolved because most state-of-the-art models treat token boundaries as incidental rather than explicit objectives Zhou et al. (2024); Zaratiana et al. (2023). A second challenge is *domain generalization*: systems trained on one discipline must adapt to new fields without extensive retraining. Supervised models often collapse out of distribution, while large language models (LLMs) with more than 10 billion parameters offer broader adaptability but are computationally expensive, making them impractical for routine KG construction. In contrast, our proposed model is lightweight, with only ∼300 million parameters. Unlike general-purpose LLMs which require billions of parameters to achieve generalization, HGNet matches the computational efficiency of specialized baselines while offering the robust zero-shot capabilities of a foundation model.

Once entities are identified, the next task is to establish edges between them, introducing two further challenges. Scientific knowledge is hierarchical, for instance, *"Deep Learning"* is a subfield of *"Machine Learning"*. Capturing such relationships requires *hierarchy-aware relation modeling* Bai et al. (2021), yet conventional models are largely *hierarchy-blind*, relying on shallow co-occurrence statistics rather than deeper conceptual structures. Beyond hierarchy, graphs must also be logically consistent: contradictions such as declaring A part of B and B part of A undermine integrity. Large language models, while capable of performing both NER and RE in a single framework, are again prohibitively expensive and yield inconsistent results on specialized, hierarchical scientific knowledge (Refer table 3 of Zhang et al. (2024). Ensuring *globally consistent structures* is therefore essential, but current methods lack mechanisms to guarantee that the graph forms a valid Directed Acyclic Graph (DAG) Chami et al. (2020). Reliable KG construction thus requires not only accurate entity recognition but also principled modeling of relational and structural dependencies.

To the best of our knowledge, we introduce the first end-to-end system designed to address all four challenges by discovering latent hierarchical structures directly from text. Our framework operates in two stages. Stage one employs *Z-NERD*, a zero-shot recognizer that ensures robust domain generalization via Orthogonal Semantic Decomposition (OSD) and captures complex entities with a Multi-Scale TCQK attention mechanism. Stage two applies *HGNet* (Hierarchy Graph Network), which builds a latent probabilistic graph, preserves hierarchical dependencies through specialized message-passing, and enforces structural integrity via two objectives: Differentiable Hierarchy Loss and Continuum Abstraction Field (CAF) Loss. To enable rigorous evaluation and mitigate data scarcity, we contribute *SPHERE*, a large-scale, multi-domain benchmark. Across datasets such as SciERC, SciER and SPHERE, our framework achieves new state-of-the-art results: average gains of 8.08% in NER and 5.99% in RE, with even larger improvements in zero-shot settings (10.76% for NER and 26.2% for RE). Collectively, these contributions establish the first principled, empirically validated solution for building robust, high-quality scientific KGs at scale. Our main contributions include:

- **Z-NERD:** We propose a novel, domain-agnostic NER model that significantly *outperforms all state-of-the-art baselines* on the most challenging scientific benchmarks (Refer table 1). Its core innovations are the *Multi-Scale TCQK* mechanism, which enables coherent recognition of multi-word entities by dedicating attention heads to n-gram patterns, and *Orthogonal Semantic Decomposition (OSD)*, a new technique for zero-shot generalization that identifies domain-invariant "semantic turn" signals.

- **Hierarchy Graph Network (HGNet):** We introduce a GNN architecture for relation extraction that establishes a *new state-of-the-art* by a significant margin on complex hierarchical benchmarks (Refer table 2, 3, 4). It learns and reasons over a latent, probabilistic conceptual graph and, unlike standard GNNs, uses *specialized message-passing channels* (parent-to-child, child-to-parent, and peer-to-peer) to preserve the directional flow of hierarchical information.

- **A Geometric Theory of Abstraction:** We introduce a novel paradigm for representing hierarchical knowledge. We are the first to formalize abstraction as an intrinsic geometric property of standard Euclidean space, realized through a learnable *Abstraction Field Vector* that creates a universal axis of generality. This approach, enforced by our *Continuum Abstraction Field (CAF)*

*Loss*, offers a more direct and interpretable alternative to complex methods like hyperbolic embeddings.

- **The SPHERE Dataset:** To address the critical bottleneck of data scarcity, we created and release *SPHERE*, the first large-scale, multi-domain benchmark specifically designed for hierarchical relation extraction. Generated via a novel methodology, it contains over 1 million paragraphs and 111,000 annotated relations, enabling more robust training and evaluation of complex KG construction models.

## 2 RELATED WORKS

The task of constructing a Knowledge Graph (KG) from scientific literature involves two primary sub-tasks: Named Entity Recognition (NER) to identify conceptual nodes, and Relation Extraction (RE) to identify the semantic edges between them. This section situates our work within existing paradigms for these tasks, highlighting the persistent gaps that motivate our proposed framework.

### 2.1 ENTITY RECOGNITION IN SCIENTIFIC TEXT

High-performance scientific NER has been dominated by supervised transformer models such as SciBERT Beltagy et al. (2019) and BioBERT Lee et al. (2019), pre-trained on large scientific corpora and fine-tuned on task-specific data. This paradigm achieves state-of-the-art performance on in-domain benchmarks and has been scaled to foundation models like BioMedLM Bolton et al. (2024), yet it faces a critical architectural limitation. The ability to capture complex, multi-word entities (e.g., "in situ transmission electron microscopy") arises only as an emergent property of contextual embeddings rather than a dedicated feature, often resulting in fragmented or incomplete recognition. Our Z-NERD framework addresses this gap through the Multi-Scale TCQK mechanism, which intrinsically modifies attention to force heads to specialize in n-gram patterns of varying lengths, offering a principled, structural solution.

A second limitation is poor domain generalization: supervised models degrade sharply on out-of-domain text. Zero-shot methods such as GLiNER Zaratiana et al. (2023) and UniversalNER Zhou et al. (2024) reformulate the task as span matching, while general-purpose LLMs like GPT-4 OpenAI (2025) show impressive but inconsistent zero-shot performance Zhang et al. (2024). Yet these approaches still depend on surface semantics or world knowledge. By contrast, our Orthogonal Semantic Decomposition (OSD) trains the model to detect domain-agnostic *semantic turns*—points where new concepts are introduced, shifting focus from vocabulary to discourse structure. This enables Z-NERD to achieve robust zero-shot performance beyond the reach of semantic matching.

### 2.2 RELATION EXTRACTION AND HIERARCHICAL MODELING

Relation extraction (RE) has evolved from localized, sentence-level models to corpus-level systems capable of multi-hop reasoning across documents. Early neural approaches relied on pipeline architectures, but error propagation soon motivated joint models that simultaneously extract entities and relations Zhong & Chen (2021); Yamada et al. (2020); Yan et al. (2023). Benchmarks such as SciERC Luan et al. (2018) and SciER Zhang et al. (2024) have been instrumental in driving progress, enabling transformer-based methods that achieve state-of-the-art performance on fine-grained scientific relations. However, these methods remain confined to sentence-level reasoning and fail to capture the long-range dependencies and cross-sentence evidence chains that are central to scientific literature.

To address this limitation, recent work has shifted toward cross-document relation extraction, employing graph neural networks (GNNs) and multi-hop retrieval to link entity mentions across documents and aggregate distributed evidence Wang et al. (2022b); Lu et al. (2023). Yet such methods typically rely on surface features like co-occurrence or syntactic proximity, conflating textual adjacency with genuine conceptual relatedness and yielding noisy graphs. Meanwhile, hierarchy-aware approaches such as hierarchical attention Han et al. (2018) and reinforcement learning frameworks Takanobu et al. (2019) show promise but are tailored to shallow taxonomies, limiting their applicability to the deep, nested, and implicit hierarchies of scientific knowledge. We therefore introduce *HGNet*, the first GNN architecture explicitly designed for hierarchical relation extraction in scientific literature. HGNet builds a latent conceptual graph and leverages parent, child, and peer message-

passing channels to model the directional flow of information, disentangling textual proximity from conceptual hierarchy and enabling the capture of both local and global dependencies while preserving the layered structure of scientific knowledge.

## 2.3 Geometric and Logical Representations of Hierarchy

A key challenge in learning hierarchical structures is ensuring they are both logically and geometrically sound. Our HGNet captures directional information flow and disentangles textual proximity from conceptual hierarchy, but still requires a principled embedding space for global consistency. To address this, we introduce a geometric perspective: instead of merely extracting relations, we learn a hierarchy representation that respects logical constraints and abstraction levels. While hyperbolic geometry is often used for low-distortion tree embeddings Nickel & Kiela (2017), our approach defines a new paradigm, learning a globally consistent abstraction ordering directly in Euclidean space. This is achieved via the Continuum Abstraction Field (CAF) Loss, which orients the embedding space along a learnable universal "axis of abstraction." Simpler and more interpretable, this prior integrates with our Differentiable Hierarchy Loss, enforcing logical constraints such as acyclicity. Together, these losses ensure the learned KG is both geometrically organized and logically coherent.

## 3 Methodology

Our framework consists of a unified, co-trained architecture utilizing a shared SciBERT encoder. First, Z-NERD processes raw scientific text to identify and extract entity mentions. Second, HGNet takes the contextualized entity embeddings from this shared encoder as input, which maintains the document-level context, and learns their hierarchical and peer relationships, constructing a globally consistent knowledge graph.

### 3.1 Z-NERD: Zero-Shot Entity Recognition

Z-NERD is an efficient tagging model that addresses two key challenges in NER: recognizing multi-word entities and generalizing to new domains. Its architecture first applies Orthogonal Semantic Decomposition to the input embeddings to extract domain-agnostic features, then feeds these enriched representations into a transformer encoder modified with our Multi-Scale TCQK mechanism.

#### 3.1.1 Domain Generalization via Orthogonal Semantic Decomposition (OSD)

To overcome domain overfitting, a model must learn to recognize abstract linguistic patterns rather than memorizing domain-specific vocabulary. This requires identifying features that are invariant across different scientific fields.

We therefore hypothesize that Hypothesis 3.1 : *robust domain generalization can be achieved by training a model to rely on features that explicitly isolate the introduction of new semantic concepts, rather than simply tracking the overall semantic flow*. By providing the model with a "semantic turn" signal (a measure of how much the meaning deviates from the preceding context), we can make it sensitive to the underlying logical structure of the text instead of overfitting to vocabulary.

We achieve this by decomposing the change vector between consecutive word embeddings, $\Delta E_t = E_{\text{text}_t} - E_{\text{text}_{t-1}}$, into two orthogonal components. The *sustaining component* is the projection of this change onto the previous word's embedding, representing elaboration. The *divergent component*, which is orthogonal to the previous word's direction, captures the introduction of a new concept.

$$v_{\text{sustaining}_t} = \frac{\Delta E_t \cdot E_{\text{text}_{t-1}}}{\|E_{\text{text}_{t-1}}\|^2} E_{\text{text}_{t-1}} \tag{1}$$

$$v_{\text{divergent}_t} = \Delta E_t - v_{\text{sustaining}_t} \tag{2}$$

We concatenate the divergent vector $v_{\text{divergent}_t}$ with the original contextual embedding $E_{\text{text}_t}$ (Refer figure 6). This enriched representation provides the model with the domain-invariant signal of conceptual shifts necessary for robust zero-shot generalization.

### 3.1.2 COHERENT MULTI-WORD ENTITIES VIA MULTI-SCALE TCQK ATTENTION

Standard self-attention mechanisms lack a strong architectural bias for word adjacency, often failing to identify the precise boundaries of long entities. This leads to fragmented predictions and an incomplete understanding of complex concepts.

Our guiding hypothesis is that Hypothesis 3.2 : *robust, variable-length entity detection can be achieved by designing a self-attention mechanism where different heads are architecturally specialized to capture n-gram patterns of different lengths*. By fusing the global reach of attention with the local sequence awareness of convolutions at multiple scales, the model can learn to recognize single tokens, short phrases, and long entities in parallel.

We introduce the Multi-Scale Temporal Convolutional Queries & Keys (TCQK) mechanism to realize this. Before computing attention scores, we modify the Query ($\mathbf{Q}$) and Key ($\mathbf{K}$) vectors using 1D convolutions. We partition the $H$ attention heads into $G$ groups, assigning each group $g$ a convolutional kernel $C_g$ with a specific size $k_g$ (e.g., 1, 3, 5). For each head $h$ in group $g$, we compute:

$$\mathbf{Q}_{\text{conv},h} = C_g(\mathbf{Q}_h); \quad \mathbf{K}_{\text{conv},h} = C_g(\mathbf{K}_h) \tag{3}$$

This modification intrinsically alters the self-attention mechanism, compelling different heads to specialize in n-gram patterns of varying lengths. This allows the model to capture both short acronyms and long chemical names as single, coherent concepts. Note: We apply Multi-Scale TCQK mechanism over the concatenated embeddings from orthogonal semantic decomposition. (Refer figure 6 for more details)

## 3.2 HGNET: HIERARCHY GRAPH NETWORK

Given the entities extracted by Z-NERD, the goal of the Hierarchy Graph Network (HGNet) is to estimate the conditional distribution $P(T_{\text{local}} \mid D, \mathcal{G}_K)$, where each local relation triplet (start entity, relation, end entity) is constrained by a global Hierarchical Knowledge Graph (HKG) $\mathcal{G}_K$. The input entities ($\boldsymbol{h}_u, \boldsymbol{h}_v$) are the contextualized output embeddings from the SciBERT encoder, ensuring the document-level context is maintained for relationship prediction, a standard procedure for efficiency in SOTA RE models. Since $\mathcal{G}_K$ is unobserved, HGNet must jointly infer its structure and leverage it for reasoning. The model is organized around three core components, each grounded in a specific hypothesis about hierarchical consistency. (For a complete visual overview of the architecture, refer Fig. 7 in the Appendix.)

### 3.2.1 PROBABILISTIC HIERARCHICAL MESSAGE PASSING

Traditional Graph Neural Networks (GNNs) are fundamentally "hierarchy-blind." They operate on a single, undifferentiated graph, propagating messages uniformly across all connections. This approach is flawed as it cannot distinguish between information flowing "up" from a specific child, "down" from an abstract parent, or "sideways" from a peer, thereby corrupting the learned representations. To address this, our work is guided by the hypothesis that Hypothesis 3.3 : *a GNN can preserve and leverage hierarchical structure if its message-passing architecture is explicitly designed to respect it*. By creating distinct, parallel channels for information flowing along different axes of the hierarchy, the model can learn specialized, context-aware update functions, leading to richer and more robust entity embeddings.

To realize this, our architecture operates on a probabilistic graph where relations are treated as learnable variables. First, a *Latent Relation Predictor* (MLP) estimates the probability distribution over relation types $\mathcal{R} = \{\text{parent-of, peer-of, no-edge}\}$ for every pair of entity nodes $(u, v)$:

$$P_{uv} = \text{softmax}(\text{MLP}([\boldsymbol{h}_u || \boldsymbol{h}_v])) \tag{4}$$

These probabilities serve as soft edge weights for a three-channel message passing scheme. For a given node $v$ at layer $k$, we compute aggregated messages, each with a separate, learnable weight matrix ($W_{\text{up}}, W_{\text{down}}, W_{\text{peer}}$) to capture the unique semantics of each relational direction:

1. **Parental (Upstream) Aggregation**: $\boldsymbol{m}_v^{\text{parents}} = \sum_{u \in V} P_{uv}^{\text{parent}} \cdot (W_{\text{up}} \boldsymbol{h}_u^{(k)})$

2. **Child (Downstream) Aggregation**: $\boldsymbol{m}_v^{\text{children}} = \sum_{u \in V} P_{vu}^{\text{parent}} \cdot (W_{\text{down}} \boldsymbol{h}_u^{(k)})$

3. **Peer Aggregation**: $\boldsymbol{m}_v^{\text{peers}} = \sum_{u \in V} P_{uv}^{\text{peer}} \cdot (W_{\text{peer}} \boldsymbol{h}_u^{(k)})$

Finally, these three context-specific messages are concatenated with the node's previous state and passed through an update MLP to produce the final, structure-aware embedding for the next layer:

$$\boldsymbol{h}_v^{(k+1)} = \text{UpdateMLP}([\boldsymbol{h}_v^{(k)} || \boldsymbol{m}_v^{\text{parents}} || \boldsymbol{m}_v^{\text{children}} || \boldsymbol{m}_v^{\text{peers}}]) \tag{5}$$

### 3.2.2 Logical Coherence via Differentiable Hierarchy Loss (DHL)

A critical challenge in learning a latent graph is that, without explicit constraints, the model has no incentive to ensure its structure is globally coherent. During training, it might predict logically impossible structures, such as cycles (e.g., A is a part of B, and B is a part of A) or shortcuts that skip hierarchical levels (e.g., mistaking a grandparent for a parent). These structural inconsistencies corrupt the message-passing process and lead to semantically invalid graphs. We therefore hypothesize that Hypothesis 3.4 : *we can enforce a logically sound latent hierarchy by explicitly and differentiably penalizing these structural impossibilities*. In particular, by introducing a composite loss that punishes cycles and invalid shortcuts, we guide the model toward a parameter space where the latent graph forms a valid Directed Acyclic Graph (DAG) with a strict parent-child hierarchy.

This is achieved with the *Differentiable Hierarchy Loss* ($\mathcal{L}_{\text{hierarchy}}$), a regularizer operating on the predicted parent-of adjacency matrix, $A_{\text{parent}}$. It is a weighted sum of two components:

$$\mathcal{L}_{\text{hierarchy}} = \lambda_{\text{acyclic}} \mathcal{L}_{\text{acyclic}} + \lambda_{\text{separation}} \mathcal{L}_{\text{separation}} \tag{6}$$

The first component is an *Acyclicity Loss*, which uses the trace of a matrix exponential to differentiably ensure the graph is a DAG (Refer appendix A.7, for accelerated calculation using Krylov's subspace) Here, $d$ is the number of nodes (entities) in the graph. For proof of why this function pushes our graph structure to be DAG, refer appendix A.11.

$$\mathcal{L}_{\text{acyclic}} = \text{tr}(e^{A_{\text{parent}} \circ A_{\text{parent}}}) - d \tag{7}$$

The second component is a *Hierarchical Separation Loss*, which penalizes shortcut edges that skip intermediate hierarchical levels (Refer appendix A.7 for efficient computation). Formally, it is defined as:

$$\mathcal{L}_{\text{separation}} = \sum_{u,w} (A_{\text{parent}}^2)_{uw} \cdot (A_{\text{parent}})_{uw} \tag{8}$$

Here, $(A_{\text{parent}}^2)_{uw}$ counts the number of length-2 paths from node $u$ to node $w$, and the elementwise product with $(A_{\text{parent}})_{uw}$ selects only direct edges that skip an intermediate node. This encourages the model to maintain a strict parent-child hierarchy by discouraging shortcuts.

### 3.2.3 Geometric Coherence via Continuum Abstraction Field (CAF) Loss

A model's embedding space is typically geometrically "flat," lacking an intrinsic structure for abstraction. While a model might learn that "RNN" and "LSTM" are related, it fails to encode that an RNN is a more general concept, leaving embeddings as a disorganized cloud of points. Our approach is founded on the hypothesis that Hypothesis 3.5 : *hierarchical understanding is a fundamental geometric property of the embedding space*. By organizing all concepts along a single, universal "axis of abstraction," the model can embed hierarchical information directly into the vector representations, making the abstraction level of a concept an intrinsic property of its learned embedding.

We introduce the *Continuum Abstraction Field (CAF) Loss* ($\mathcal{L}_{\text{caf}}$) to impose this geometric structure. It introduces a *learnable unit vector*, the *Abstraction Field Vector $\boldsymbol{w}_{\text{abs}}$*, that defines this universal axis (Refer appendix A.5 for more details on the unit abstraction field vector). An entity $v$'s abstraction score is defined as its projection onto this axis: $\hat{y}_{\text{abs}}(v) = \boldsymbol{h}_v \cdot \boldsymbol{w}_{\text{abs}}$. This abstraction score is a continuous, real-valued number, ensuring the model learns a fluid continuum rather than a limited

number of fixed, discrete levels. The composite loss, $\mathcal{L}_{\text{caf}} = \mathcal{L}_{\text{ranking}} + \gamma_1 \mathcal{L}_{\text{anchor}} + \gamma_2 \mathcal{L}_{\text{regression}}$, shapes this structure using three distinct objectives:

- **Ranking Component:** Enforces relative parent-child ordering with a margin $\delta$.

$$\mathcal{L}_{\text{ranking}} = \frac{1}{|\mathcal{E}_{\text{part-of}}|} \sum_{(c,p) \in \mathcal{E}_{\text{part-of}}} \max(0, (\boldsymbol{h}_c - \boldsymbol{h}_p) \cdot \boldsymbol{w}_{\text{abs}} + \delta) \tag{9}$$

- **Anchoring Component:** Pins known root ($\mathcal{V}_s$) and leaf ($\mathcal{V}_t$) nodes to scores of 1 and 0.

$$\mathcal{L}_{\text{anchor}} = \frac{1}{|\mathcal{V}_s|} \sum_{v_s \in \mathcal{V}_s} (\boldsymbol{h}_{v_s} \cdot \boldsymbol{w}_{\text{abs}} - 1)^2 + \frac{1}{|\mathcal{V}_t|} \sum_{v_t \in \mathcal{V}_t} (\boldsymbol{h}_{v_t} \cdot \boldsymbol{w}_{\text{abs}} - 0)^2 \tag{10}$$

- **Regression Component:** Pulls predicted scores towards ground-truth topological depth scores $y_{\text{topo}}(v)$, which are derived for all benchmarks by performing a topological sort on the ground truth hierarchical relations.

$$\mathcal{L}_{\text{regression}} = \frac{1}{|\mathcal{V}_{\text{train}}|} \sum_{v \in \mathcal{V}_{\text{train}}} ((\boldsymbol{h}_v \cdot \boldsymbol{w}_{\text{abs}}) - y_{\text{topo}}(v))^2 \tag{11}$$

Note that while the margin $\delta$ in $\mathcal{L}_{\text{ranking}}$ could theoretically limit the number of discrete levels to $1/\delta$, this constraint is effectively relaxed by $\mathcal{L}_{\text{regression}}$, which acts as the dominant global anchor pulling each embedding toward its true topological depth. The model thus learns a continuous spectrum of abstraction rather than discrete levels (see Appendix A.6 for empirical evidence).

This transforms abstraction from a simple regression target into an organizing principle of the entire embedding space.

### 3.2.4 FINAL RELATION PREDICTION

The relations {parent-of, peer-of, no-edge} described in Section 3.2.1 are an internal mechanism used solely for structure regulation. The $\mathbf{h}^{(k+1)}$ embedding produced by HGNet represents the final, optimized structure-aware representation. The extraction of the actual, task-specific ⟨head, relation, tail⟩ triplets is then performed by a standard downstream classification head (the same type utilized by models such as HGERE or PL-Marker) operating on this refined representation. This head takes the structure-aware $\mathbf{h}^{(k+1)}$ embedding as input and predicts the full set of fine-grained relations required by the benchmarks. The loss from this external task, $\mathcal{L}_{\text{RE}}$, constitutes the primary task objective of the entire framework.

### 3.2.5 COHERENT ARCHITECTURE AND JOINT OPTIMIZATION

While Sections 3.2.1–3.2.4 define the modular components of HGNet, the system operates as a single, unified framework, where all elements are simultaneously optimized in one end-to-end forward pass. This co-training mechanism ensures the learned structure is globally consistent, logically sound, and geometrically coherent. The entity embeddings ($\boldsymbol{h}_u, \boldsymbol{h}_v$) are the contextualized outputs from the shared SciBERT encoder of the Z-NERD stage. The *Latent Relation Predictor* estimates the initial probability distribution $P_{uv}$ over relations, which immediately initiates two parallel paths: Logical Regularization and Message Passing. The predicted parent matrix ($\boldsymbol{A}_{\text{parent}}$) feeds directly into the Differentiable Hierarchy Loss ($\mathcal{L}_{\text{hierarchy}}$), which penalizes structural errors like cycles ($\mathcal{L}_{\text{acyclic}}$) and shortcut edges ($\mathcal{L}_{\text{separation}}$). Concurrently, the probabilities $P_{uv}$ are used as soft edge weights to guide the three-channel Probabilistic Message Passing GNN, which produces the enhanced, structure-aware entity embeddings ($\boldsymbol{h}_v^{(k+1)}$).

These final embeddings $\boldsymbol{h}_v^{(k+1)}$ are then used to compute the Continuum Abstraction Field (CAF) Loss ($\mathcal{L}_{\text{caf}}$). This loss enforces geometric ordering, shaping the embedding space along the universal axis of abstraction. The embeddings are also passed to a final classification head, which predicts the task-specific ⟨head, relation, tail⟩ triplets ($\mathcal{L}_{\text{RE}}$). The total loss for the model is a weighted composite sum of the primary task objective and the two structural regularizers:

$$\mathcal{L}_{\text{Total}} = \mathcal{L}_{\text{RE}} + \lambda_1 \mathcal{L}_{\text{hierarchy}} + \lambda_2 \mathcal{L}_{\text{caf}} \tag{12}$$

This joint optimization is the core of HGNet: $\mathcal{L}_{\text{hierarchy}}$ forces the graph structure to be logically sound, while $\mathcal{L}_{\text{caf}}$ forces the node embeddings to be geometrically sound.

### VALIDATION OF STRUCTURAL LOSSES

The efficacy of enforcing structural and geometric coherence is confirmed through targeted ablation studies. Ablating the Differentiable Hierarchy Loss (DHL) led to a notable drop in performance, confirming the necessity of penalizing logical inconsistencies like cycles and shortcut edges. Similarly, removing the Continuum Abstraction Field (CAF) Loss resulted in a significant degradation in Rel+ F1 score, validating that embedding generality as an intrinsic geometric property is critical for hierarchical reasoning. (For detailed empirical results, refer to the section 4.)

Refer figure 7 for detailed visualization of all components of HGNet. Refer appendix A.12 for a short proof sketch of HGNet as a generalized attention mechanism. For a short discussion on convergence of HGNet, refer appendix A.13

## 4    EXPERIMENTS

In this section, we present a comprehensive empirical evaluation of our proposed Z-NERD and HGNet frameworks. We first detail the experimental setup, then present the main performance results against strong baselines, and finally conduct a series of ablation studies and analyses to validate our core hypotheses.

### 4.1    EXPERIMENTAL SETUP

**Datasets**    We evaluate our models on a diverse set of scientific information extraction benchmarks. This includes four established datasets: SciERC Luan et al. (2018), SciER Zhang et al. (2024), BioRED, and SemEval-2017 Task 10 Augenstein et al. (2017). These datasets span multiple scientific domains, feature complex entity and relation types, and are standard benchmarks for this task. For fair comparison we report all the metrics on the Out of Distribution official test sets. To address the scarcity of large-scale annotated data, we also introduce SPHERE, a new, large-scale dataset created via a novel LLM-based generate-and-annotate methodology. SPHERE contains four distinct scientific domains (Computer Science, Physics, Biology, and Material Science), enabling robust evaluation of both in-domain and zero-shot performance.

**Evaluation Metrics**    For Named Entity Recognition (NER), we report the standard micro F1 score. For the more complex end-to-end Relation Extraction (RE) task, we use the strict Rel+ F1 metric Zhong & Chen (2021), which requires the model to correctly predict the boundaries and types for both entities in a relation, as well as the relation type itself.

**Baselines**    Our frameworks are benchmarked against a comprehensive suite of strong models. For NER, we compare Z-NERD against state-of-the-art supervised models (SciBERT, PL-Marker, HGERE), a powerful specialized model (UniversalNER-7b), and several general-purpose LLMs in a zero-shot setting. For RE, we compare HGNet against top-performing end-to-end supervised models (PL-Marker, HGERE), standard GNN architectures (GCN, GAT), and LLMs.(Refer table 1 for references). Additional experiments comparing HGNet against **Hyperbolic Baselines** and **Few-Shot CoT LLMs** are detailed in A.8 and A.9, respectively.

For implementation details, hyperparameters and SPHERE dataset generation process, refer appendices A.2 and A.3.1.

### 4.2    MAIN RESULTS

**Z-NERD for Entity Recognition**    As shown in Table 1, our Z-NERD framework sets a new state-of-the-art across all benchmark datasets, achieving an 8.08% average F1 improvement over previous supervised models. The gains are even higher in the zero-shot SPHERE domains, with a 10.76% average improvement. In contrast, general-purpose LLMs evaluated directly in zero-shot mode without task-specific fine tuning failed to produce meaningful results, mainly due to difficulties in

identifying multi-word entity boundaries. These LLMs are also much larger, highlighting Z-NERD's efficiency at under 1B parameters.

**HGNet for Relation Extraction**   The central goal of HGNet is to learn a globally coherent representation of scientific knowledge that respects its inherent hierarchical structure. We divide the relations in each dataset into two classes, hierarchical and peer, and report the macro F1 for these two classes separately. As shown in Table 2, 3 and 4, HGNet consistently outperforms all baseline models, with an average improvement of **5.99%** on the benchmark datasets and **26.20%** on the zero-shot SPHERE dataset. This demonstrates a distinct advantage on datasets characterized by complex hierarchical relations, driven by its hierarchy-aware multi-channel message-passing architecture.

### 4.3   ABLATION STUDIES AND ANALYSIS

**Analysis of Z-NERD Architecture**   To validate our architectural contributions to Z-NERD, we performed targeted ablation studies. First, removing the Multi-Scale TCQK mechanism results in a severe degradation of performance across every dataset. This sharp decline confirms Hypothesis 3.1 , validating that standard attention mechanisms are ill-equipped to handle the coherent identification of complex, multi-word entities and that an explicit architectural bias for n-gram patterns is fundamental to success. Second, removing the features from Orthogonal Semantic Decomposition (OSD) also leads to a consistent drop in F1 scores. The true significance of this component becomes most apparent in the zero-shot domain generalization task, where the performance drop is particularly pronounced. This provides compelling evidence for  Hypothesis 3.2 , confirming that isolating "semantic turns" is the key to learning abstract, domain-agnostic patterns for robust generalization. (Refer table 1) For evidence of how Orthogonal Semantic Decomposition affects the learned embeddings to improve zero-shot inference, refer appendix A.4.

**Analysis of HGNet Architecture**   The superior performance of HGNet is driven by its unique design, which we validate through ablations. The model's overall strong performance across all datasets, particularly those with deep hierarchies, provides strong empirical support for  Hypothesis 3.3 . This confirms that an explicitly hierarchy-aware GNN architecture with specialized parent, child, and peer message-passing channels produces richer and more accurate entity representations than standard GNNs. Furthermore, removing the Continuum Abstraction Field (CAF) Loss resulted in a significant degradation in Rel+ F1 score, validating  Hypothesis 3.5  by demonstrating that embedding generality as an intrinsic geometric property of the space is critical for hierarchical reasoning. Similarly, ablating the Differentiable Hierarchy Loss also led to a notable drop in performance, which confirms  Hypothesis 3.4  and underscores the necessity of enforcing logical constraints like acyclicity. (Refer table 2 and 3) For learned abstraction score analysis and qualitative error analysis, refer appendices A.10 and A.6.

Table 1: F1 scores (%) of different models on NER benchmarks. SPHERE: CS, Physics, Bio, MS report both supervised (Sup) and zero-shot (ZS) results. OSD refers to orthogonal semantic decomposition. Z-NERD w/o OSD and w/o TCQK refer to ablations.

| Models | SciERC | SciER | BioRED | SemEval | CS | | Physics | | Bio | | MS | |
|---|---|---|---|---|---|---|---|---|---|---|---|---|
| | | | | | Sup | ZS | Sup | ZS | Sup | ZS | Sup | ZS |
| *Supervised Baselines* | | | | | | | | | | | | |
| SciBERT Ye et al. (2022) | 67.52 | 70.71 | 89.15 | 49.14 | 68.19 | 57.02 | 72.90 | 61.22 | 75.83 | 68.45 | 67.29 | 57.14 |
| PL-Marker Yan et al. (2023) | 70.32 | 74.04 | 86.41 | 47.69 | 68.64 | 56.39 | 72.83 | 60.51 | 75.78 | 66.17 | 66.72 | 57.92 |
| HGERE Yan et al. (2023) | 75.92 | 81.19 | 89.43 | 48.25 | 69.82 | 58.95 | 72.46 | 60.67 | 76.42 | 68.51 | 67.24 | 58.03 |
| UniversalNER-7b Zhou et al. (2024) | 66.09 | 73.13 | 88.46 | 47.60 | | | | | OOM | | | |
| *Zero-Shot LLM Baselines* | | | | | | | | | | | | |
| llama-3.3-70b Touvron et al. (2023) | 46.20 | 49.57 | 54.82 | 30.16 | | | | | OOM | | | |
| qwen3-32b Qwen et al. (2025) | 41.63 | 46.52 | 31.71 | 26.48 | | | | | OOM | | | |
| llama-3.1-8b-instant Touvron et al. (2023) | 33.96 | 31.21 | 33.58 | 21.70 | | | | | OOM | | | |
| *Proposed Approach (Z-NERD)* | | | | | | | | | | | | |
| Z-NERD w/o TCQK | 73.43 | 75.12 | 84.43 | 47.85 | 68.47 | 59.35 | 74.92 | 61.74 | 73.92 | 68.30 | 69.48 | 57.73 |
| Z-NERD w/o OSD | 74.39 | 80.27 | 90.12 | 50.98 | 76.93 | 62.04 | 76.68 | 65.17 | 82.40 | 73.29 | 78.24 | 63.45 |
| **Z-NERD** | **78.84** | **82.71** | **91.05** | **52.26** | **80.47** | **69.52** | **82.39** | **73.19** | **84.35** | **74.21** | **83.96** | **72.28** |

## 5   CONCLUSION AND FUTURE WORK

We present a novel two-stage framework for automated knowledge graph construction in the scientific domain. The first stage, Z-NERD, combines Orthogonal Semantic Decomposition with a Multi-Scale TCQK attention mechanism for robust, domain-agnostic recognition of complex entities. The second stage, HGNet, employs a probabilistic graphical model with specialized message-

Table 2: Rel+ F1 scores (%) of different models on SciERC, SciER, BioRED, and SemEval 2017 for two relation types (*Hierarchical* and *Peer*), along with overall F1 across all relation types. For BioRED, which only has *Peer* relations, the overall score equals the peer score. The same entity prediction method is used across models for fair comparison. All values are rounded to two decimals.

| Models | SciERC | | | SciER | | | BioRED | SemEval 2017 | | |
|---|---|---|---|---|---|---|---|---|---|---|
| | Hier. | Peer | Overall | Hier. | Peer | Overall | Overall | Hier. | Peer | Overall |
| *Supervised Models* | | | | | | | | | | |
| PL-Marker Ye et al. (2022) | 35.60 | 44.97 | 41.63 | 40.25 | 61.84 | 56.78 | 29.87 | 32.96 | 43.40 | 37.19 |
| HGERE Yan et al. (2023) | 37.72 | 47.35 | 43.86 | 43.79 | 64.35 | 58.47 | 32.39 | 33.81 | 45.73 | 38.63 |
| PURE Zhong & Chen (2021) | 34.39 | 38.46 | 36.78 | 38.53 | 56.21 | 49.35 | 29.41 | 28.94 | 41.35 | 34.92 |
| *Zero-Shot LLM Models* | | | | | | | | | | |
| GPT-3.5 Turbo Ye et al. (2023) | 14.97 | 15.02 | 14.98 | 8.35 | 8.91 | 8.58 | 6.36 | 16.30 | 17.13 | 16.74 |
| openai/gpt-oss-120b Ye et al. (2023) | 19.68 | 21.27 | 20.45 | 27.93 | 27.52 | 27.64 | 7.15 | 23.59 | 24.16 | 23.88 |
| llama-3.3-70b-versatile Touvron et al. (2023) | 22.15 | 22.53 | 22.39 | 23.97 | 25.06 | 24.59 | 7.29 | 23.65 | 25.38 | 24.12 |
| qwen/qwen3-32b Qwen et al. (2025) | 16.57 | 19.33 | 18.20 | 24.02 | 24.45 | 24.28 | 6.71 | 20.92 | 21.38 | 21.09 |
| llama-3.1-8b-instant Touvron et al. (2023) | 13.30 | 14.27 | 13.92 | 17.15 | 17.69 | 17.43 | 5.48 | 14.11 | 14.46 | 14.24 |
| *Supervised GNN-based Models* | | | | | | | | | | |
| GCN | 40.13 | 48.78 | 45.62 | 47.37 | 63.89 | 57.35 | 31.93 | 34.08 | 45.92 | 38.96 |
| GCN w/o $\mathcal{L}_{DHL}$ | 38.46 | 48.51 | 44.98 | 46.85 | 64.22 | 56.89 | 32.28 | 32.82 | 45.72 | 37.99 |
| GAT | 40.37 | 49.11 | 46.21 | 47.35 | 64.29 | 57.64 | 32.40 | 34.47 | 46.19 | 39.25 |
| GAT w/o $\mathcal{L}_{DHL}$ | 38.96 | 49.25 | 45.48 | 47.03 | 64.23 | 57.30 | 32.74 | 33.52 | 45.88 | 38.43 |
| *Proposed Approaches* | | | | | | | | | | |
| HGNet w/o $\mathcal{L}_{DHL}$ | 48.52 | 55.37 | 51.68 | 59.10 | 65.95 | 62.79 | 34.31 | 42.16 | 49.42 | 45.05 |
| HGNet w/o $\mathcal{L}_{CAF\ Loss}$ | 42.70 | 52.14 | 47.33 | 54.75 | 61.21 | 58.67 | 33.09 | 38.58 | 43.28 | 41.19 |
| HGNet | 50.96 | 55.41 | 53.19 | 62.36 | 67.02 | 65.38 | 33.85 | 45.37 | 50.64 | 47.03 |

Table 3: Rel+ F1 scores (%) on SPHERE dataset variants (Computer Science, Physics, Biology, Material Science) for two relation types (*Hier.*, *Peer*). We also report overall F1 across all relation types. *All models use the same entity prediction method for a fair comparison.* Subscripts indicate improvement over SOTA model HGERE.

| Models | Comp. Sci. | | | Physics | | | Biology | | | Mat. Sci. | | |
|---|---|---|---|---|---|---|---|---|---|---|---|---|
| | Hier. | Peer | All | Hier. | Peer | All | Hier. | Peer | All | Hier. | Peer | All |
| *Supervised Models* | | | | | | | | | | | | |
| PL-Marker Ye et al. (2022) | 51.98 | 57.04 | 55.29 | 50.22 | 56.48 | 53.51 | 52.35 | 53.76 | 53.03 | 52.96 | 53.27 | 53.12 |
| HGERE Yan et al. (2023) | 54.20 | 59.86 | 57.93 | 53.17 | 58.90 | 56.28 | 54.52 | 56.47 | 55.21 | 55.84 | 55.86 | 55.43 |
| *Proposed Approaches* | | | | | | | | | | | | |
| HGNet (ours) | **77.40** | **81.36** | **79.51** | **76.93** | **83.47** | **80.60** | **82.53** | **84.29** | **83.74** | **81.91** | **85.64** | **83.65** |
| w/o $\mathcal{L}_{DHL}$ | 73.62 | 74.83 | 74.17 | 74.01 | 75.30 | 74.66 | 79.15 | 78.64 | 78.90 | 77.43 | 76.92 | 77.28 |
| w/o $\mathcal{L}_{CAF}$ | 67.14 | 65.89 | 66.50 | 64.51 | 66.24 | 65.96 | 75.17 | 73.29 | 74.13 | 75.95 | 77.38 | 76.32 |

Table 4: Zero-shot Rel+ F1 (%) when trained on Physics+Biology and evaluated on Comp. Sci. and Mat. Sci. datasets. Due to the expensive nature of these experiments, we only tested our zero-shot performance on the best performing state-of-the-art models in our previous experiments.

| Models | Comp. Sci. | | | Mat. Sci. | | |
|---|---|---|---|---|---|---|
| | Hier. | Peer | All | Hier. | Peer | All |
| PL-Marker Ye et al. (2022) | 28.72 | 28.41 | 28.56 | 33.10 | 34.22 | 33.85 |
| HGERE Yan et al. (2023) | 29.93 | 29.63 | 29.81 | 36.27 | 39.41 | 37.97 |
| HGNet (ours) | **59.36** | **64.07** | **62.60** | **69.92** | **71.33** | **70.62** |

passing channels, regularized by Differentiable Hierarchy and Continuum Abstraction Field losses. The latter introduces a learnable Abstraction Field Vector, ensuring logical coherence and geometric structuring around a universal abstraction axis. We also introduce SPHERE, a large-scale benchmark for scientific KG construction. Experiments show gains of up to 10.76% for NER and 26.2% for RE in zero-shot scenarios, validating our hypotheses and the efficacy of a structurally-aware approach to knowledge extraction.

Future work could extend this framework to incorporate multimodal information from figures and tables, and explore its application in dynamic, continuously updated knowledge graphs that reflect the real-time evolution of scientific fields. Additionally, syntactic filtering based on dependency parsing could be integrated as a preprocessing step to prune unlikely entity pairs, further enhancing relation extraction precision (Joshi & Rekik, 2025). Furthermore, leveraging these structured KGs for downstream reasoning tasks, such as automated hypothesis generation, presents an exciting avenue for further research.

## REPRODUCIBILITY STATEMENT

To ensure the reproducibility of our results, all source code for the Z-NERD and HGNet models and the newly introduced SPHERE dataset are publicly available at `https://github.com/basiralab/HGNet`. We have provided comprehensive details of our experimental setup, including datasets, evaluation metrics (Section 4.1), implementation, software/hardware configurations, and training hyperparameters (Appendix A.2). The methodology for generating the SPHERE dataset is further detailed in Appendix A.3.1, and all baseline models are described in Section 4.1 to facilitate fair comparison.

## ACKNOWLEDGEMENTS

We are grateful to the Computing Support Group (CSG) at Imperial College London for managing the GPU cluster used in our experiments. We also thank the anonymous reviewers for their constructive feedback, which significantly strengthened this manuscript.

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

# A  APPENDIX

## A.1  STATEMENT ON THE USE OF LARGE LANGUAGE MODELS (LLMS)

In adherence to the ICLR 2026 policy, we disclose the use of Large Language Models (LLMs) in the preparation of this manuscript and in our research methodology.

**1. Role in Dataset Generation**   As detailed in Appendix A.3.1, LLMs (specifically, a mixture of models from the GPT and Gemini families) were a core component of our research. They were programmatically used to generate and self-annotate the SPHERE dataset, which was crucial for training and evaluating our proposed models. The entire process, from KG scaffolding to sentence generation and annotation, was designed and supervised by the authors to ensure the quality and validity of the dataset.

**2. Role in Manuscript Preparation**   Beyond their role in the research itself, LLMs were also utilized as tools to aid in the preparation of this paper in the following ways:

- **Writing and Polishing:** We used LLMs (e.g., GPT-4) as advanced writing assistants. Their use was primarily focused on improving the clarity, precision, and readability of the text. This included tasks such as rephrasing sentences for better flow, correcting grammatical errors, ensuring consistent terminology, and polishing the overall prose. The core scientific ideas, arguments, and the structure of the paper were conceived and written entirely by the authors.

- **Literature Retrieval and Discovery:** LLMs were used as a supplementary tool to augment our traditional literature review process. We used them to summarize abstracts of known papers and to help identify potential related work based on keyword and concept queries. This assisted in broadening our search, but the final selection, critical reading, analysis, and citation of all literature were performed by the authors to ensure academic rigor.

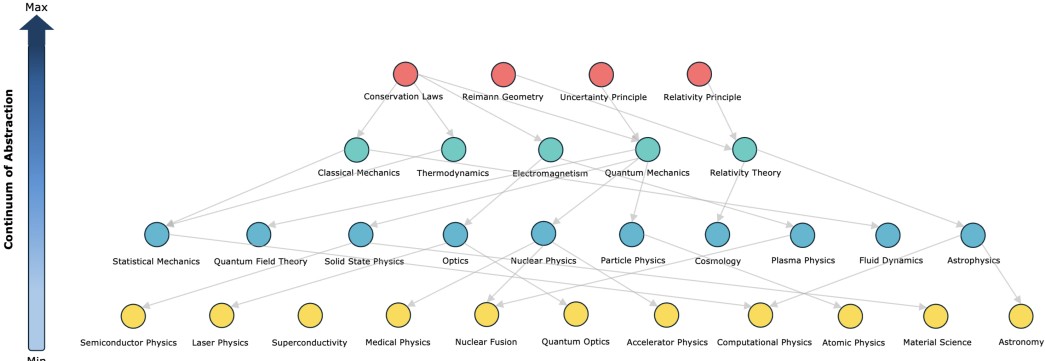

Figure 1: Continuous axis of abstraction for topics in physics.

## A.2  IMPLEMENTATION DETAILS

**Hardware and Software**   All experiments were conducted on a high-performance computing cluster equipped with NVIDIA A30 24GB GPUs. Our frameworks were implemented using PyTorch 2.1 and the Hugging Face Transformers library. For baseline models, we used their official public implementations and recommended hyperparameters to ensure fair comparison.

**Training Hyperparameters**   To ensure reproducibility, we detail the key hyperparameters for our proposed models in Table 5. We used the AdamW optimizer for all training runs and employed a linear learning rate scheduler with a warm-up phase. The optimal hyperparameters were determined via a grid search on the validation sets of the respective datasets.

Table 5: Key hyperparameters for Z-NERD and HGNet.

| Hyperparameter | Z-NERD | HGNet |
|---|---|---|
| Encoder Base Model | SciBERT-base | SciBERT-base |
| Learning Rate | $2 \times 10^{-5}$ | $1 \times 10^{-5}$ |
| Batch Size | 16 | 8 |
| Optimizer | AdamW | AdamW |
| Dropout Rate | 0.1 | 0.2 |
| Max Sequence Length | 512 | 512 |
| TCQK Kernel Sizes | [1, 3, 5, 7] | N/A |
| HGNet Layers | N/A | 3 |
| CAF Loss Margin ($\delta$) | N/A | 0.5 |
| CAF Weights ($\gamma_1, \gamma_2$) | N/A | (1.0, 0.5) |
| DHL Weights ($\lambda_{\text{acyclic}}, \lambda_{\text{separation}}$) | N/A | (1.0, 0.1) |

## A.3 SPHERE DATASET

### A.3.1 GENERATION METHODOLOGY

The SPHERE (Scientific Multidomain Large Entity and Relation Extraction) dataset was created to overcome the critical bottleneck of data scarcity in scientific RE. We employed a novel, three-phase generate-and-annotate methodology driven by a Large Language Model (in our case, mixture of GPT (OpenAI) and Gemini (Google DeepMind) models).

1. **Phase 1: Programmatic KG Scaffolding.** We first constructed a ground-truth knowledge graph to serve as a structured backbone. This was done by prompting the LLM with a high-level field (e.g., "Computer Science") and asking it to recursively expand it into more granular, interconnected sub-fields, methods, and concepts. This foundational step produced a deep and logically consistent taxonomy of over 40,000 entities across four domains before any text was generated.

2. **Phase 2: High-Throughput Sentence Generation.** With the KG as a scaffold, a high-throughput pipeline generated annotated sentences. This involved sampling small, contextually related sets of concepts from the graph (e.g., a parent, child, and peer concept) and prompting the LLM, acting as an expert technical writer, to compose a long, complex, academic-style paragraph describing their relationships.

3. **Phase 3: LLM Self-Annotation.** The newly created sentences were immediately passed back to the same LLM for self-annotation within the original context. The model performed Named Entity Recognition and Relation Extraction, linking the identified concepts back to their permanent IDs in the ground-truth KG. We observed that the LLM's annotation performance is drastically higher on text it has generated itself, enabling the creation of a large-scale (10,000 documents, 111,000 relations), high-quality corpus.

### A.3.2 STRUCTURAL COMPLEXITY AND SCALE ANALYSIS

To validate the necessity of SPHERE as a foundation benchmark, we compare its structural properties against existing gold-standard datasets in Table 6.

**Scale and Diversity.** Existing benchmarks like SciERC and BioRED are constrained by the high cost of human annotation, typically limited to roughly 500 abstracts and a single domain. In contrast, SPHERE leverages the generative scaffolding approach to scale to 10,000 documents across four distinct domains (Computer Science, Physics, Biology, Material Science). This scale is critical for pre-training "foundation" extraction models that can generalize zero-shot.

**Taxonomic Depth.** Most standard datasets utilize "flat" entity ontologies (e.g., broad categories like *Method* or *Material*). SPHERE, being generated from a deep Knowledge Graph scaffold, contains nested hierarchical definitions (e.g., *Adam Optimizer → Stochastic Optimization → Optimiza-*

*tion Method*). This distinct structural depth forces models to learn fine-grained hierarchical reasoning (tested via HGNet) rather than simple surface-level pattern matching.

**Structural Consistency and Global Scope.** A critical distinction of SPHERE is the scope of its graph topology. Standard benchmarks like SciERC are annotated at the document level, meaning the hierarchical relationships are locally inferred and often inconsistent (e.g., an entity may be a root in one document but a leaf in another). In contrast, SPHERE is generated from a Global Knowledge Graph Scaffold containing over 40,000 entities. This ensures that the hierarchical position of a concept remains globally consistent across the entire corpus, preventing the "inflated structure" or hallucinated loops often associated with unconstrained LLM generation.

Table 6: Comparison of SPHERE against standard scientific IE benchmarks. A key distinction is Graph Scope: standard datasets define hierarchies locally within isolated documents (fragmented), whereas SPHERE is generated from a single Global Knowledge Graph, ensuring hierarchical consistency across the entire corpus.

| Dataset | Domain | Docs | Relations | Graph Scope | Hierarchy Source |
|---|---|---|---|---|---|
| SciERC | CS (AI) | 500 | ~4.6k | Local (Doc-Level) | Inferred from Text |
| BioRED | Biomed | 600 | ~38k | Local (Doc-Level) | Inferred from Text |
| SciER | CS | 106 | ~12k | Local (Doc-Level) | Inferred from Text |
| **SPHERE** | **4 Domains** | **10,000** | **111,000** | **Global (Corpus-Level)** | **Pre-defined Scaffold** |

The fidelity of the SPHERE dataset is evidenced by its surprising zero-shot efficacy. When trained only on SPHERE, our model generalizes to the human-annotated SciERC and SciER benchmarks with scores of 46.55% and 59.17% respectively, outperforming the previous fully supervised state-of-the-art (HGERE). This confirms that SPHERE faithfully models the complex entity-relation dependencies of scientific text, validating our constrained generation pipeline.

Table 7: Performance comparison on SciERC and SciER datasets.

| Metric | Training Source | SciERC (Test) | SciER (Test) |
|---|---|---|---|
| **HGERE Yan et al. (2023)** | Full Supervised Training | 43.86% | 56.28% |
| **HGERE (Zero-shot transfer)** | SPHERE-CS Training Only | 25.62% | 28.34% |
| **HGNet (Zero-shot transfer)** | **SPHERE-CS Training Only** | **46.55%** | **59.17%** |

**Manual Quality Audit.** To quantitatively assess the fidelity of the SPHERE dataset and ensure minimal hallucination, we conducted a manual verification study on a randomly sampled subset of the corpus. We analyzed 1,000 entity spans and 500 relation triples against the ground-truth topological scaffold. The audit yielded an entity precision of 96.5% (measuring correct boundary and type) and a relation precision of 94.2% (measuring correct edge classification). These high precision scores confirm that our constrained "generate-from-graph" pipeline effectively enforces structural consistency while maintaining textual fluency.

### A.4 Visual Evidence for Orthogonal Semantic Decomposition

To further validate the premise of OSD, Figures 2 illustrate the average Orthogonal Semantic Velocity Norm for tokens at entity boundaries versus non-entity tokens. The plots provide compelling visual support for Hypothesis 3.2 . A clear and substantial gap emerges between the high velocity norms of boundary tokens and the low norms of non-entity tokens. This demonstrates that our engineered feature effectively captures the sharp "semantic turns" that occur when a new concept is introduced, providing a robust, domain-agnostic indicator of entity boundaries.

### A.5 Geometric Realization via an Abstraction Field

Instead of treating abstraction score as an external label to be predicted, our central hypothesis is that the abstraction score should be an *intrinsic geometric property* of the learned embedding space

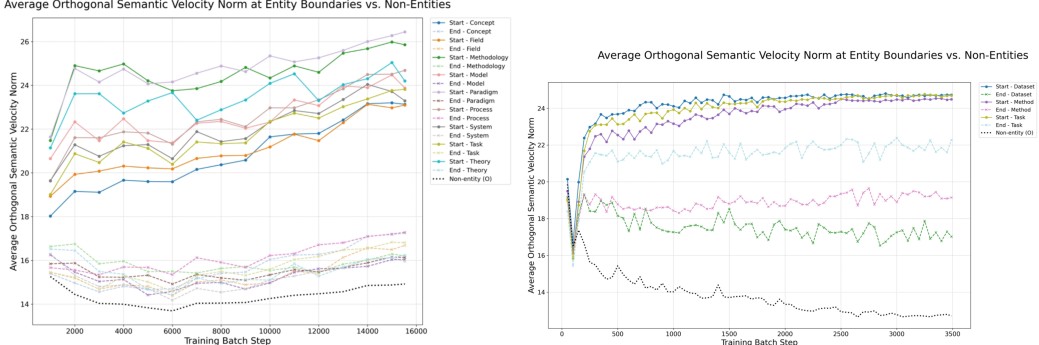

Figure 2: Average Orthogonal Semantic Velocity for tokens at entity boundaries ('Start'/'End') vs. 'Non-entity' tokens for SPHERE-CS (left) and SciER (right). The clear separation provides visual evidence for Hypothesis 3.1.

itself. We propose that the entire high-dimensional space can be oriented along a single, universal direction that represents a continuum from specificity to generality. We formalize this concept as the *Abstraction Field Vector*.

**Definition A.1 (Abstraction Field Vector)** *a learnable unit vector* $\boldsymbol{w}_{abs} \in \mathbb{R}^d$. *This vector defines the primary axis of abstraction within the embedding space. The predicted abstraction score,* $\hat{y}_{abs}(v)$, *for any concept* $v$ *with embedding* $\boldsymbol{h}_v$ *is then simply its orthogonal projection onto this vector:*

$$\hat{y}_{abs}(v) := \boldsymbol{h}_v \cdot \boldsymbol{w}_{abs} \tag{13}$$

**Justification for a Single Universal Axis:** The choice to model abstraction with a single unit vector is a deliberate application of simplicity and a method for imposing a strong, beneficial inductive bias. While one could model abstraction using multiple orthogonal vectors or a more complex non-linear function, such approaches would implicitly assume the existence of multiple, independent "types" of abstraction. Our formulation, by contrast, hypothesizes that the dominant organizing principle of a scientific knowledge hierarchy is a single, primary dimension of generality versus specificity. This constraint forces the model to discover the most salient and universal axis of abstraction that is consistent across all entities, rather than overfitting to spurious, domain-specific hierarchical patterns. This mirrors findings in other areas of representation learning, where simple linear axes have been shown to capture profound semantic relationships (e.g., the famous 'king - man + woman' analogy in word embeddings). By reducing abstraction to a single, interpretable dimension, we ensure the learned geometric structure is not only robust but also directly analyzable. The empirical success of this method across multiple domains serves as strong validation for this simplifying, yet powerful, geometric assumption.

This formulation is powerful because it transforms the abstract notion of "generality" into a concrete, measurable geometric arrangement. A concept's position along this axis directly reflects its level of abstraction. This approach ensures that the learned hierarchy is not an afterthought but the primary organizing principle of the entire embedding space, making the learned representations globally coherent and interpretable. Refer figure 1 for visualization of continuum of abstraction in physics domain.

### A.6 LEARNED ABSTRACTION SCORE ANALYSIS

To qualitatively assess the geometric structure learned by HGNet, we visualized the distribution of the final abstraction scores for entities within each domain of the SPHERE dataset, as shown in Figure 3. Based on the programmatic, recursive generation of the underlying knowledge graph, the ideal distribution would exhibit an exponential decay, with a high density of concrete entities at low abstraction scores and a progressively smaller number of entities at higher levels of abstraction. The analysis reveals distinct, domain-specific patterns that reflect the inherent structure learned from each field.

The Computer Science domain (d) aligns most closely with this expected pattern, showing a clear concentration of entities at lower abstraction values and a long tail of increasingly abstract concepts. In contrast, the Material Science data (a) shows a distribution heavily clustered at lower scores, while the Biology data (c) displays a more gradual decline, likely reflecting a flatter hierarchy in its source text. It is crucial to note, however, that even the Computer Science distribution is not a perfect match for the ground-truth hierarchy. The visible deviations from an ideal curve highlight that some concepts are still misplaced along the abstraction axis. These imperfections in the learned geometric structure are precisely what lead to a non-perfect Rel+ F1 score, highlighting the tight coupling between representational geometry and task performance.

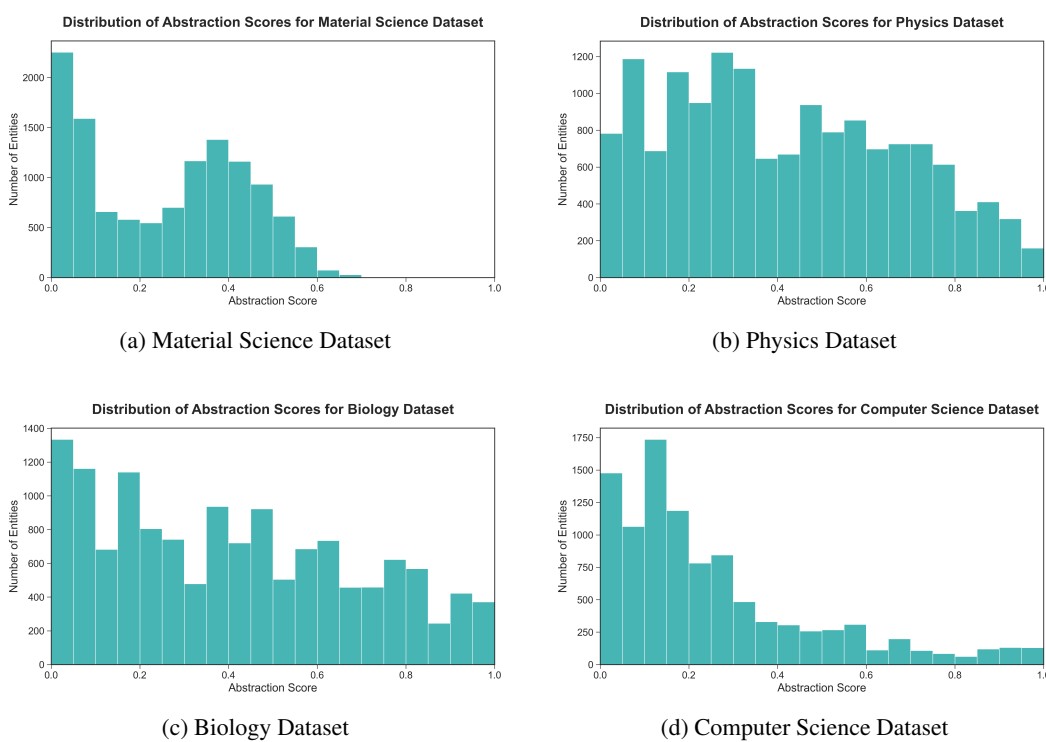

(a) Material Science Dataset

(b) Physics Dataset

(c) Biology Dataset

(d) Computer Science Dataset

Figure 3: Distribution of abstraction scores across different scientific domains, showing distinct patterns for each field.

## A.7 SCALABLE ACYCLICITY REGULARIZATION VIA KRYLOV SUBSPACE METHODS

A potential computational bottleneck in our framework is the *Differentiable Hierarchy Loss* (Eq. 7), which involves the calculation of a matrix exponential. For a graph with $n$ entities, the parent-of adjacency matrix $\mathbf{A}_{\text{parent}}$ is of size $n \times n$. A direct computation of the matrix exponential, $\exp(\mathbf{A}_{\text{parent}} \circ \mathbf{A}_{\text{parent}})$, scales with a time complexity of $\mathcal{O}(n^3)$, which can become prohibitive for the large-scale knowledge graphs targeted by our work.

To ensure the scalability of our approach, this term can be efficiently approximated using Krylov subspace methods (Saad, 2003). Instead of explicitly forming the dense $n \times n$ matrix exponential, these iterative methods approximate its action on a vector by projecting the matrix onto a low-dimensional Krylov subspace, $\mathcal{K}_m$, of dimension $m \ll n$.

The computational cost of this approach is dominated by two steps. First, the construction of an orthonormal basis for the subspace, typically via the Arnoldi iteration, requires $m$ matrix-vector products. Since the learned adjacency matrix $\mathbf{A}_{\text{parent}}$ is inherently sparse, with a number of non-zero entries denoted by $\text{nnz}(\mathbf{A}_{\text{parent}})$, this step has a complexity of $\mathcal{O}(m \cdot \text{nnz}(\mathbf{A}_{\text{parent}}))$. Second, the exponential of the small $m \times m$ projected matrix is computed directly, which incurs a cost of $\mathcal{O}(m^3)$.

Therefore, the total time complexity for the approximation is $\mathcal{O}(m \cdot \text{nnz}(\mathbf{A}_{\text{parent}}) + m^3)$. Furthermore, to compute the trace required by our loss function, Krylov methods is combined with stochastic trace estimators, Hutchinson method (Avron & Toledo, 2011), to approximate $\text{tr}(\exp(\mathbf{A}))$ without ever forming the full matrix.

**Hierarchical Separation Loss** The second component, the *Hierarchical Separation Loss* (Eq. 8), is defined as $\mathcal{L}_{\text{separation}} = \sum_{u,w}(\mathbf{A}_{\text{parent}}^2)_{uw} \cdot (\mathbf{A}_{\text{parent}})_{uw}$. A direct computation would first involve squaring the matrix $\mathbf{A}_{\text{parent}}$, an operation that, even for sparse matrices, can be costly as the resulting matrix $\mathbf{A}_{\text{parent}}^2$ may be significantly denser. However, we can reinterpret this loss as a sum over specific graph structures. The term $(\mathbf{A}_{\text{parent}}^2)_{uw}$ represents the sum of weights of all paths of length two from entity $u$ to $w$. The loss thus penalizes the existence of a direct "shortcut" edge $(u, w)$ when such two-step paths exist. This structure allows for a far more efficient calculation. Instead of matrix multiplication, we can compute the sum by iterating through all 2-paths in the graph. An efficient algorithm involves iterating through each node $v$ and considering all pairs of its incoming edges $(u, v)$ and outgoing edges $(v, w)$. For each such 2-path $u \rightarrow v \rightarrow w$, we perform a sparse lookup to check for the existence of the direct edge $(u, w)$. The total complexity of this approach is approximately $\mathcal{O}(\sum_{v \in V} \text{in-degree}(v) \cdot \text{out-degree}(v))$, which is directly proportional to the local sparsity of the graph and avoids the costly formation of $\mathbf{A}_{\text{parent}}^2$.

This analysis demonstrates that both components of the Differentiable Hierarchy Loss can be computed efficiently, ensuring that the enforcement of a globally consistent DAG structure remains computationally feasible even for knowledge graphs containing thousands of entities.

## A.8 EXTENDED GEOMETRIC BASELINE ANALYSIS

To validate the efficacy of the Continuum Abstraction Field (CAF) against non-Euclidean approaches, we compare HGNet against two strong geometric baselines using the same SciBERT backbone:

- **HGCN (Hyperbolic GCN)** Chami et al. (2019): Maps embeddings to the Poincaré ball manifold, theoretically optimized for hierarchical trees.
- **Order-Embeddings** Vendrov et al. (2015): Enforces partial order constraints via cone geometry ($E = ||\max(0, v - u)||^2$).

Table 8 presents the results. HGNet outperforms both baselines. We observe that HGCN requires extensive tuning of the Riemannian Adam optimizer and often struggles with "Peer" relations that violate strict tree geometries, whereas HGNet's Euclidean CAF objective remains stable and accurate.

Table 8: Comparison against Geometric Baselines (Rel+ F1 %). HGNet outperforms non-Euclidean methods on both standard and large-scale hierarchical datasets.

| Model | Geometry | SciERC (Overall) | SPHERE-CS (Overall) |
|---|---|---|---|
| HGCN Chami et al. (2019) | Hyperbolic ($\mathbb{D}^n$) | 45.82 | 66.35 |
| Order-Embeddings Vendrov et al. (2015) | Cone ($\mathbb{R}_+^n$) | 44.27 | 67.97 |
| **HGNet (Ours)** | **Euclidean + CAF** | **53.19** | **79.51** |

## A.9 FEW-SHOT LLM EVALUATION

To ensure a fair comparison regarding reasoning capabilities, we evaluated Llama-3-8B using a 3-Shot Chain-of-Thought (CoT) strategy. We provided the model with three context-response pairs demonstrating step-by-step relation extraction before querying the target sentence.

As shown in Table 9, while CoT provides a notable performance boost over the zero-shot setting (+5.73% on SciERC), the model still significantly underperforms compared to HGNet. Qualitative error analysis reveals that while CoT helps identifying relation types, the LLM continues to struggle with precise entity boundary detection (e.g., including determiners or punctuation in the span), which is penalized by the strict Rel+ metric.

Table 9: Impact of Prompting Strategy on Llama-3-8B Performance (Rel+ F1 %).

| Model | Prompting Strategy | SciERC | SciER |
|---|---|---|---|
| Llama-3-8B | Zero-Shot | 13.72 | 14.95 |
| | 3-Shot CoT | 19.45 | 25.18 |
| **HGNet** | **Supervised** | **53.19** | **62.36** |

## A.10 QUALITATIVE ERROR ANALYSIS

CORRECTED ERROR: PREVENTING HIERARCHICAL SHORTCUTS

A significant advantage of HGNet is its ability to maintain a strict, multi-level hierarchy by penalizing "shortcut" edges that skip intermediate levels. This corrects errors where a local model might conflate a grandparent relationship with a direct parent one. Consider a biology paper discussing genetics:

- Sentence 1: The *SRY* gene is responsible for encoding the Testis-determining factor protein.
- Sentence 2: A conserved motif within the Testis-determining factor protein is the High-mobility group (HMG) box, which binds to DNA.

From these sentences, a correct hierarchy is established: (HMG box $\rightarrow$ Testis-determining factor protein $\rightarrow$ SRY gene). However, another sentence might state: "The DNA-binding function of the *SRY* gene is conferred by its HMG box." A local model, seeing this direct functional link, could incorrectly infer a direct compositional relation: (HMG box, Part-Of, SRY gene). This creates a flawed, flattened hierarchy.

**HGNet corrects this error.** Its *Hierarchical Separation Loss* ($\mathcal{L}_{\text{separation}}$) is explicitly designed to prevent this. Once the model identifies the valid two-step path from "HMG box" to "SRY gene", the loss function penalizes the prediction of a direct edge between them. This forces the model to respect the intermediate entity ("Testis-determining factor protein"), ensuring the final graph accurately reflects the nested biological structure.

**Robustness to Non-Hierarchical Structures.** Here, we explain how HGNet behaves when the underlying structure is not a strict tree (e.g., multiple inheritance or cross-links). We observe that the *Peer* message-passing channel is critical in these scenarios. In cases of multiple inheritance (e.g., "Reinforcement Learning" being a child of both "Machine Learning" and "Control Theory"), HGNet successfully assigns high probability to both parent edges because the DAG constraint ($\mathcal{L}_{acyclic}$) permits multiple parents, only forbidding cycles. However, we note a failure mode in "loopy" citations where definitions are circular (A defines B, B defines A). In such rare cases, the acyclicity loss forces the model to arbitrarily break the loop, potentially dropping a valid semantic link.

## A.11 JUSTIFICATION FOR THE DIFFERENTIABLE ACYCLICITY LOSS

To enforce a Directed Acyclic Graph (DAG) structure, we require a differentiable function that penalizes the presence of cycles within the graph represented by the learned adjacency matrix $\mathbf{A}_{\text{parent}}$. Our loss function is built upon a well-established connection between the algebraic properties of a graph's adjacency matrix and its topological structure.

The foundation of this approach lies in the observation that the number of distinct walks of length $k$ from a node $i$ to a node $j$ is given by the entry $(i, j)$ of the matrix power $\mathbf{A}^k$. Consequently, a cycle, which is a walk from a node back to itself, is captured by the diagonal entries. The sum of these diagonal elements, or the trace $\text{tr}(\mathbf{A}^k)$, therefore counts the total number of cycles of length $k$ across the entire graph.

A graph is a DAG if and only if it contains no cycles of any length $k \geq 1$, which implies that $\text{tr}(\mathbf{A}^k) = 0$ for all $k \geq 1$. To aggregate this condition over all possible cycle lengths into a single, smooth function, we leverage the matrix exponential, defined by its Taylor series

$\exp(\mathbf{A}) = \sum_{k=0}^{\infty} \frac{1}{k!} \mathbf{A}^k$. Due to the linearity of the trace operator, we have:

$$\text{tr}(\exp(\mathbf{A})) = \sum_{k=0}^{\infty} \frac{\text{tr}(\mathbf{A}^k)}{k!} = \text{tr}(\mathbf{I}) + \text{tr}(\mathbf{A}) + \frac{\text{tr}(\mathbf{A}^2)}{2!} + \dots$$

For a graph with $d$ nodes that is a perfect DAG, all trace terms for $k \geq 1$ vanish, causing the expression to simplify elegantly to $\text{tr}(\exp(\mathbf{A})) = \text{tr}(\mathbf{I}) = d$.

Based on this property, our loss function, $\mathcal{L}_{\text{acyclic}} = \text{tr}(\exp(\mathbf{A}_{\text{parent}})) - d$, is formulated. This objective function is non-negative and equals zero only when the graph is perfectly acyclic. By minimizing this loss during training, a computation made efficient by modern numerical libraries such as PyTorch's 'torch.linalg.matrix_exp', we guide the model to learn an adjacency matrix $\mathbf{A}_{\text{parent}}$ whose corresponding graph structure satisfies the DAG constraint.

## A.12 Perspective: HGNet as a Generalized Attention Mechanism

At its core, the self-attention mechanism, which powers modern Transformers, can be understood as a form of message passing on a fully connected graph. Each token in a sequence acts as a node, and it updates its representation by aggregating information from every other token. This is very powerful, as it allows the model to capture long-range dependencies. However, it is also a brute-force approach. It operates under the assumption that any token could be relevant to any other, leading to two major limitations:

1. **Computational Inefficiency:** The number of connections grows quadratically with the sequence length, making it computationally expensive for long documents.

2. **Semantic Noise:** In a scientific document, the relationship between the vast majority of token pairs is meaningless. Forcing a token like "LSTM" to attend to every instance of "the" or "is" introduces significant noise and forces the model to expend capacity learning to ignore these irrelevant connections.

The fundamental insight of our work is that we can create a far more powerful and efficient reasoning mechanism by moving from a dense, token-level graph to a sparse, *entity-level graph*. Instead of every word attending to every other word, we want key scientific concepts to attend only to other relevant scientific concepts. By "skipping the middle tokens" and operating directly on the meaningful entities, we can focus the model's capacity on learning the true global structure of knowledge. Our Hierarchical GNN is the formal embodiment of this principle, representing a more advanced and generalized form of attention.

### Proof Sketch: From Full Attention to Structured, Hierarchical Attention

To prove this, let us first formulate the standard self-attention mechanism in the language of Graph Neural Networks.

**1. Self-Attention as a GNN on a Fully Connected Graph** The update rule for a single token embedding $\boldsymbol{h}_i$ in a self-attention layer is:

$$\boldsymbol{h}_i' = \sum_{j \in \mathcal{V}_{\text{all}}} \alpha_{ij}(\boldsymbol{h}_j W_V) \tag{14}$$

where $\mathcal{V}_{\text{all}}$ is the set of *all* tokens in the sequence, and $\alpha_{ij}$ is the attention weight between token $i$ and token $j$. This is precisely a GNN message-passing step where the graph is **fully connected**, meaning every token is a neighbor of every other token. The message from node $j$ to node $i$ is its transformed value, $\boldsymbol{m}_{j \to i} = \boldsymbol{h}_j W_V$, and the aggregation is a weighted sum, with attention scores serving as the weights. This is a powerful but unstructured mechanism. It treats all potential connections as equally plausible *a priori*.

**2. The Hierarchical GNN as a Generalized, Structured Attention** Our Hierarchical GNN introduces a powerful inductive bias by replacing the fully connected graph with a sparse, semantically

meaningful graph, one based on the learned hierarchy. The update rule for an entity embedding $h_v$ is:

$$h_v^{(k+1)} = \text{UpdateMLP}([h_v^{(k)}|m_v^{\text{parents}}|m_v^{\text{children}}|m_v^{\text{peers}}]) \quad (15)$$

Let's analyze one of these components, the message from parents:

$$m_v^{\text{parents}} = \sum_{u \in \mathcal{N}_v^{\text{parents}}} \alpha_{vu}^{\text{parent}}(W_{\text{parent}} h_u^{(k)}) \quad (16)$$

This is also an attention mechanism, but with three crucial generalizations. First, through *Graph Sparsification*, the aggregation is no longer over all possible nodes $\mathcal{V}_{\text{all}}$ but is instead over a small, semantically relevant subset, $\mathcal{N}_v^{\text{parents}}$. This prunes the vast majority of noisy, irrelevant connections, focusing the model's attention on the relationships that truly matter and directly addressing both the computational and semantic noise problems. Second, instead of a single, monolithic attention mechanism, our GNN employs *Multi-Channel Attention* with multiple, specialized channels. It learns separate projection matrices ($W_{\text{parent}}, W_{\text{child}}, W_{\text{peer}}$) and attention mechanisms for each type of hierarchical relationship, allowing the model to learn different "types" of attention. For example, learning to "inherit" abstract properties from parents while "aggregating" specific evidence from children. Third, through *Entity-Level Reasoning*, the nodes in our graph are not tokens but aggregated entity concepts representing stable ideas across the entire corpus. This provides a much more robust and global context for reasoning than the ephemeral, document-specific context of individual tokens.

**Conclusion of Proof**    The standard self-attention mechanism is a special case of our Hierarchical GNN framework under a specific set of simplifying assumptions. These assumptions are that the graph is fully connected ($\mathcal{N}_v = \mathcal{V}_{\text{all}}$ for all $v$), that there is only one message-passing channel (e.g., only a "peer" channel), and that the nodes represent tokens, not global entities. By relaxing these assumptions, our Hierarchical GNN generalizes the attention mechanism to operate on a sparse, structured, multi-channel graph of global concepts. This is not merely an incremental improvement; it is a fundamental shift from brute-force pattern matching to structured, hierarchical reasoning. It allows the model to capture the kind of radial, layered knowledge depicted in the conceptual image 4 of the scientific domain, making it a far more powerful and efficient architecture for understanding complex, interconnected information.

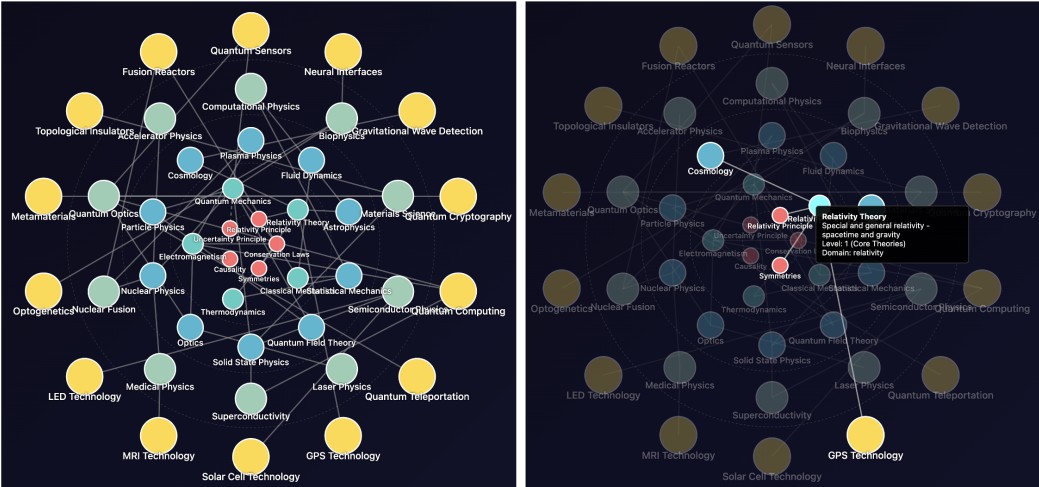

Figure 4: Example of physics domain hierarchical knowledge graph. Hierarchy radially extends outward.

### A.13    DISCUSSION ON CONVERGENCE OF HGNET

Given that HGNet is a probabilistic model, it's important to understand why its predicted probabilities converge toward a consistent graph structure rather than fluctuating randomly. The primary

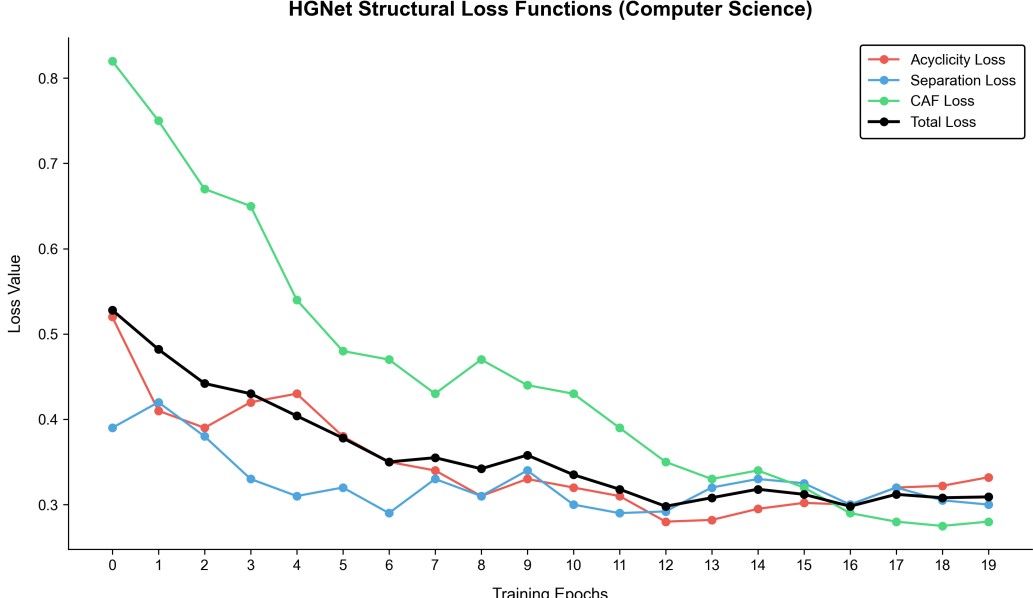

Figure 5: Loss plot of HGNet on computer science SPHERE dataset.

reason is that the model is not initialized from scratch. Instead, it uses a standard MLP classifier to generate the initial edge probabilities between entities.

As demonstrated by the baseline models in our experiments, which rely on an MLP for classification, these approaches are reasonably effective, often achieving Rel+ F1 scores exceeding 30-40% on their own. By using this as a starting point, HGNet's probabilistic message-passing begins with a well-informed "draft" of the graph. This process is far more efficient than random initialization; it's like solving a jigsaw puzzle where a significant portion of the pieces are already in their approximate correct locations, allowing the model to focus on refining the details rather than building the entire structure from scratch. Refer loss plot 5.

### A.14    COMPUTATIONAL COMPLEXITY AND EFFICIENCY ANALYSIS

We conduct a comprehensive analysis of parameter efficiency, computational cost (FLOPs), and inference throughput to validate our lightweight claims.

**Efficiency vs. Generalization Landscape.**    Table 10 benchmarks HGNet against General-purpose LLMs, Specialized SOTA methods (PL-Marker, HGERE), and lightweight Graph Neural Networks (GCN, GAT). HGNet occupies a unique "sweet spot": it matches the generalization of LLMs while maintaining the throughput of specialized models.

**Component-Wise Parameter Breakdown.**    Table 11 details the parameter distribution of the full HGNet pipeline. We employ a two-stage architecture (Z-NERD and HGNet) where decoupling implies the worst case parameter setting. Notably, the Z-NERD stage is architecturally heavier (42.4M trainable params) due to the Multi-Scale TCQK mechanism, which employs 8 parallel convolutional heads with wide projection matrices ($d_{proj} = 2048$) to capture dense n-gram contexts. The HGNet stage utilizes a lighter, structure-aware GNN (31.6M trainable params) to reason over the sparse entity graph.

**Training Overhead.**    Structural losses (DHL/CAF) are not computed during inference. During training, the Krylov subspace approximation reduces the exact matrix exponential calculation time from ∼150ms to ∼12ms per batch, rendering the overhead negligible ($< 5\%$ total training time).

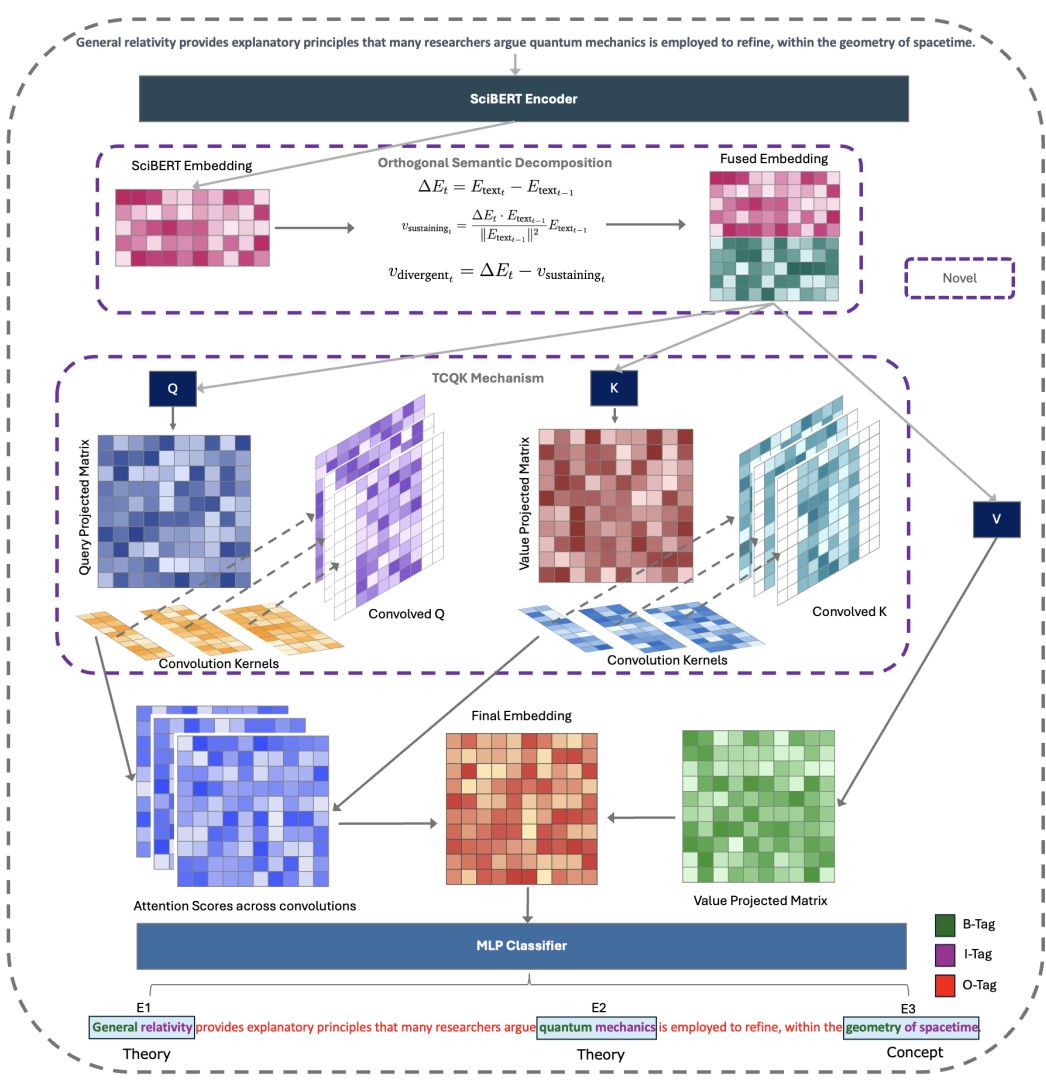

Figure 6: Main figure explaining the proposed Z-NERD algorithm. For TCQK, multi-head for each convolution has been shown as single head for simplicity. B refers to begin entity, I refers to inside entity and O refers to outside entity.

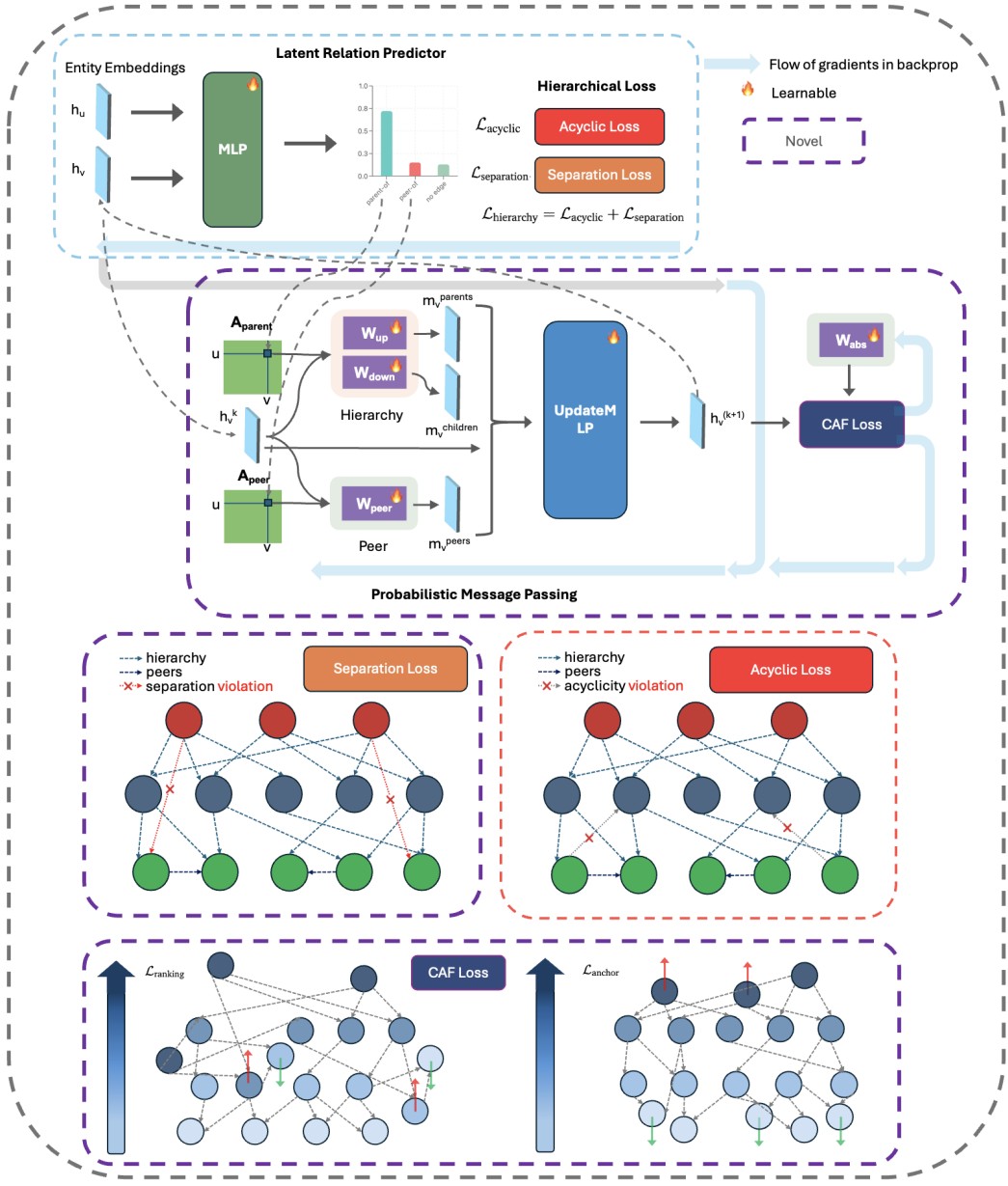

Figure 7: Main figure of proposed HGNet illustrating all proposed components. For clarity, we omit $\mathcal{L}_{\text{regression}}$, since it is simply a regression loss applied over the graph topology, similar in nature to standard losses such as mean squared error or binary cross entropy.

Table 10: Efficiency Landscape on SciERC (A30 GPU, Batch=8). GFLOPs are estimated per input instance. HGNet (Full Pipeline) provides superior throughput compared to pipeline-based SOTA models despite robust parameter capacity.

| Model | Params | GFLOPs | Speed (doc/s) | Mem (GB) | Zero-Shot Gen. |
|---|---|---|---|---|---|
| *Large Language Models* | | | | | |
| Llama-3-70B | ∼70B | >140k | ∼0.5 | OOM | High |
| Llama-3-8B | ∼8B | >16k | ∼4.2 | 16.0+ | Moderate |
| *Specialized SOTA* | | | | | |
| PL-Marker | ∼220M | 44.0 | 12.4 | 7.2 | Low |
| HGERE | ∼220M | 22.5 | 14.1 | 9.5 | Low |
| *Graph Baselines* | | | | | |
| SciBERT+GCN | ∼110M | 22.0 | 48.2 | 6.1 | Low |
| SciBERT+GAT | ∼110M | 22.1 | 46.8 | 6.3 | Low |
| *Proposed* | | | | | |
| **HGNet** | **∼293M** | **44.7** | **14.6** | **10.5** | **High** |

Table 11: Detailed parameter breakdown of the HGNet framework (Two-Stage Configuration, worst case parameters). The Z-NERD stage incorporates high-capacity TCQK attention (8 Heads) to resolve complex multi-word boundaries, while HGNet utilizes specialized message-passing layers.

| Stage | Component | Params (M) | % of Total |
|---|---|---|---|
| **Stage 1: Z-NERD** | Specialized SciBERT Encoder | 109.5 | 37.4% |
| | Multi-Scale TCQK (8 Heads) | 42.4 | 14.5% |
| **Stage 2: HGNet** | Specialized SciBERT Encoder | 109.5 | 37.4% |
| | Hierarchical GNN Layers | 31.6 | 10.7% |
| **Total** | **Full Two-Stage Pipeline** | **∼293.0** | **100%** |

