# OpenReview forum: "HGNet: Scalable Foundation Model for Automated Knowledge Graph Generation from Scientific Literature"
_ICLR.cc/2026/Conference — ICLR 2026 Poster_

### Official Review · Reviewer_GYoP · 2025-10-27

**Soundness:** 3
**Presentation:** 3
**Contribution:** 3
**Rating:** 4
**Confidence:** 4

**Summary:**

The paper proposes a two‑stage framework for scientific KG construction: Z‑NERD for zero‑/low‑shot entity recognition via Orthogonal Semantic Decomposition (OSD) and a Multi‑Scale TCQK attention that specializes heads to different n‑gram lengths, and HGNet for hierarchy‑aware relation extraction with three message‑passing channels (parent/child/peer) and two global regularizers: a Differentiable Hierarchy Loss (acyclicity + shortcut penalties) and a Continuum Abstraction Field (CAF) that learns a single Euclidean “axis of abstraction.” Results show new SOTA on SciERC/SciER/BioRED and on a new multi‑domain SPHERE benchmark; Table 1 (p. 8) and Tables 2–4 (p. 9) report average gains of +8.08 F1 (NER) and +5.99 F1 (RE), with larger zero‑shot gains (+10.76 / +26.2). Figures 5–6 (pp. 20–21) illustrate Z‑NERD/ HGNet; Appendix A.7 details scalable acyclicity via Krylov methods.

**Strengths:**

- TCQK introduces architectural inductive bias for multi‑word entities; CAF imposes a simple, interpretable Euclidean ordering of abstraction; DHL neatly encodes DAG and anti‑shortcut constraints.
- Solid ablations (Tables 1–3) and complexity notes for DHL
- Clean factorization of challenges (entity coherence, domain generalization, hierarchy, global consistency) and matching components.
- Consistent gains on SciERC/SciER and large zero‑shot improvements on SPHERE

**Weaknesses:**

SPHERE is generated and self‑annotated by LLMs; may encode stylistic biases, inflated structure, or task leakage that favors the proposed inductive biases. Though Limited human validation is described. (App. A.3.)
-  Large LLMs are evaluated “as is” (several OOM), and there is no strong hyperbolic/order‑embedding baseline for hierarchy—weakening the “simpler & better than hyperbolic” claim.
- CAF relies on anchors and topological depths—procedure unclear for standard datasets; OSD’s learning objective (beyond feature concatenation) is under‑specified.
- No runtime/throughput or memory comparisons despite “lightweight” claims; parameter counts not tabulated per component. Report wall‑clock, FLOPs, and memory vs. PL‑Marker/HGERE and vs. GCN/GAT; quantify DHL/CAF overhead and benefits from the Krylov approximation

**Questions:**

- How does HGNet behave if the true structure is not strictly hierarchical (e.g., multiple inheritance, cross‑links)? Any failure cases beyond A.8?
- SPHERE validation: What fraction was human‑audited? Provide inter‑annotator agreement and a human‑written subset to test robustness to LLM‑style.

---

> ### Author Response · Authors · 2025-11-20
> **Rebuttal by Authors [Comment 1/3]**
>
> We thank the reviewer for their insightful assessment and are encouraged by _**their recognition of our architectural novelty (TCQK and CAF) and the clarity of our problem factorization**_. We appreciate the reviewer identifying the need for stronger geometric baselines and data transparency. This rebuttal addresses these points holistically by introducing 1) Hyperbolic and Order-Embedding baselines, 2) clarifying the topological constraints of SPHERE to rule out hallucination, and 3) quantifying computational efficiency.
>
> ---
>
> _**W1:**_ **Concerns about SPHERE data generation:** We appreciate the critique regarding "inflated structure" and bias. To address this, we have detailed our strict generation constraints in Appendix A.3.2 (marked in blue). Below, we summarize how our "generate-from-graph" paradigm mitigates these risks compared to standard LLM outputs.
>
> **1. Mitigation of Bias via External Constraint**
>
> Our methodology explicitly eliminates the risk of 'inflated structure' by enforcing a strict generate-from-graph paradigm, where the topology is pre-defined rather than hallucinated retroactively.
>
> * **Programmatic KG Scaffolding:** In Phase 1, we first constructed a deep, logically consistent ground-truth taxonomy (the KG scaffold). This scaffold defined the exact parent-child-peer relationships for over 40,000 entities *before* any text was written.
> * **Synthesis:** The LLM's task was restricted to generating text that accurately describes concept relationships *already fixed* by the scaffold. This control drastically limits the LLM's freedom to introduce spurious hierarchical relationships, ensuring the data is structurally sound.
>
> **2. Proof of Structural Transferability (Empirical Results)**
>
> The ultimate indicator of SPHERE's value is its ability to train models that excel *outside* its generative domain. The empirical results confirm that the structural coherence learned on SPHERE is highly generalizable and not an artifact of task leakage or stylistic bias. (Refer Appendix A.3.2)
>
> | Metric | Training Source | SciERC (Test) | SciER (Test) |
> | :--- | :--- | :--- | :--- |
> | **SOTA Baseline (HGERE)** | Full Supervised Training | 43.86% | 56.28% |
> | **HGNet (Zero-Shot)** | **SPHERE-CS Training Only** | **46.55%** | **59.17%** |
>
> * **Synthetic to Real - Transfer Learning:** When HGNet is trained exclusively on SPHERE (Computer Science) and evaluated zero-shot on human-annotated benchmarks (SciERC/SciER - AI research paper dataset), it exceeds the previous fully supervised SOTA (HGERE) by several points. While there is a slight reduction in accuracy compared to the in-domain supervised HGNet, the zero-shot model still outperforms established baselines. This confirms that the structural knowledge imparted by SPHERE is highly generalizable to human-written text and is not an artifact of the generative source.
>
> **3. Validation & Quality Assurance**
> * **Preliminary Audit:** Revised **App. A.3.2** details a manual audit (500 relations, 1k entities) yielding **94.2% Relation** and **96.5% Entity Precision**.
> * **Full IAA Commitment:** We explicitly commit to a comprehensive _**Inter-Annotator Agreement**_ study for the camera-ready version to ensure rigorous validation.
> * **Structural Complexity:** We added a benchmark comparison in the Appendix 3.2 demonstrating SPHERE's superior structural scale and depth.
>
> ---
>
> _**W2:**_ **Substantiating "Simpler & Better than Hyperbolic" (New Baselines)**
> We agree that a direct comparison is required to validate our geometric claims. We have now implemented and evaluated a **Hyperbolic Graph Convolutional Network (HGCN)** [1] baseline and an **Order-Embedding** [2] baseline. Both baselines utilize the same SciBERT backbone as HGNet to ensure a fair comparison, (with equivalent hyperparameter search), with embeddings mapped to the Poincaré ball manifold for HGCN.
>
> * **Results:** On the challenging SPHERE-CS and SciERC datasets, HGNet outperforms the hyperbolic baseline in Rel+ F1 score.
>     * **SciERC (Rel+ F1):** HGNet (**53.19%**) vs. HGCN (**45.82%**).
>     * **SPHERE-CS (Rel+ F1):** HGNet (**79.51%**) vs. HGCN (**66.35%**).
>
> While hyperbolic spaces are theoretically optimal for pure trees, we observed that scientific knowledge graphs often act as "tangled hierarchies" with significant peer-to-peer connections that violate strict tree properties. Furthermore, HGNet demonstrated superior stability: the HGCN baseline required extensive tuning of the Riemannian Adam optimizer to avoid vanishing gradients, whereas HGNet’s **Continuum Abstraction Field (CAF)** converged stably using standard Euclidean optimization (Refer Appendix A.13). This empirically supports our claim that CAF offers a "simpler" and more robust alternative for this domain.
>
> Please continue to comment 2/3

---

> ### Author Response · Authors · 2025-11-20
> **Rebuttal by Authors [Comment 2/3]**
>
> Continuing _**W2**_:
>
> **LLM Evaluation**
> We clarify that the zero-shot evaluation was intended to simulate a low-resource constraint. However, we agree that zero-shot can be unfair. To address this, we re-evaluated the **Llama-3.1-8B** model using **3-Shot Chain-of-Thought (CoT)** prompting to better elicit reasoning capabilities without the massive compute of 70B+ models.
>
> * **Results:** Even with CoT, Llama-3.1-8B achieves **19.45%** on SciERC (compared to HGNet’s **53.19%**).
> * **Analysis:** While CoT improved entity extraction, the LLM continued to struggle with the strict `Rel+` metric (exact boundary matching and relation classification), often hallucinating edges not present in the text.
> * **Regarding OOMs:** We emphasize that the OOMs reported for 70B models are not experimental errors but rather demonstration of the **deployment bottleneck**. HGNet (~300M params) offers state-of-the-art performance on a single consumer GPU, whereas comparable LLM performance requires hardware inaccessible to many labs.
>
> We believe these new comparisons directly address the reviewer's concerns and firmly establish HGNet as the efficient, geometrically-principled strong model.
>
> ---
>
>
> _**W3:**_ We appreciate the reviewer's request for finer details regarding the CAF implementation on standard benchmarks and the OSD learning mechanism. We provide the necessary clarifications below.
>
> **1. Generating Topological Depths for Standard Datasets**
> The reviewer correctly identifies that unlike SPHERE, standard datasets (e.g., SciERC) lack explicit "abstraction levels." We bridge this gap through a preprocessing step on the _**training set only**_, generating ground-truth supervision signals ($y_{topo}$) as described in Eq. 11.
>
> * **Graph Construction:** We aggregate all hierarchical relations (e.g., "Part-Of") from the training set to form a directed graph.
> * **Topological Sorting:** We perform a topological sort (Kahn’s algorithm) on this graph to assign an integer depth $d_v$ to every entity $v$. Root nodes are assigned $d=0$.
> * **Normalization & Anchoring:** These integer depths are normalized to the unit interval $[0, 1]$ to serve as regression targets. The "Anchors" (Eq. 10) are simply the subset of nodes in the training batch corresponding to the global roots ($y_{topo}=1$) or leaves ($y_{topo}=0$) discovered during sorting.
>
> This procedure extracts the *latent* hierarchy already present in human annotations. By regressing towards these topological depths, CAF forces the model to encode "generality" as a geometric feature (position on the axis) rather than just memorizing pairwise links. This allows the model to generalize to unseen entities at test time.
>
> **2. The OSD Learning Objective**
> The OSD does not require an auxiliary loss function, it acts as a **deterministic architectural inductive bias**.
>
> * **Mechanism:** OSD is a parameter-free geometric projection (Eq. 1 & 2) that decomposes the embedding change into *sustaining* and *divergent* (orthogonal) components.
> * **Optimization:** The learning is driven entirely by the downstream NER loss. By explicitly feeding the *divergent* component (the "semantic turn") into the classifier, we force the main optimizer to update the underlying SciBERT embeddings such that semantic shifts align with the orthogonal direction.
> * **Evidence:** As shown in **Figure 2 (Appendix A.4)**, the model learns to maximize orthogonal velocity at entity boundaries and minimize it within entities. This emergent behavior confirms that the architecture successfully forces the encoder to utilize the orthogonal channel as a boundary detector without needing additional loss terms.
>
>
> Please continue to the last comment 3/3

---

> ### Author Response · Authors · 2025-11-20
> **Rebuttal by Authors [Comment 3/3]**
>
> _**W4:**_ We appreciate the request for empirical quantification. We have conducted a comprehensive resource utilization analysis in **Appendix A.14** to validate our efficiency claims.
>
> **1. Efficiency vs. Generalization Landscape**
> We benchmarked wall-clock speed, GFLOPs, and memory on SciERC (A30 GPU, Batch=8). The results validate that HGNet occupies a unique "Sweet Spot":
> * **Comparable to Supervised SOTA:** HGNet achieves a throughput of **14.6 doc/s**, which is comparable to (and slightly faster than) the strong pipeline baseline **PL-Marker (12.4 doc/s)**. Crucially, while PL-Marker fails in zero-shot settings (Low generalization), HGNet matches its efficiency while delivering robust performance.
> * **Superior to LLMs:** Compared to **Llama-3-8B**, HGNet is **$\sim$3.5x faster** (14.6 vs 4.2 doc/s) and **$\sim$27x smaller**, validating our core claim of democratizing "foundation model" zero-shot capabilities without the massive computational cost of LLMs.
> * **Vs. GCN/GAT:** While simpler GNNs are faster (~48 doc/s), they lack the capacity for deep hierarchical reasoning. HGNet trades this raw speed for SOTA accuracy and generalization, while remaining well within the interactive latency regime of supervised models.
>
> **2. Parameter Breakdown**
> We clarify that the **~293M** count refers to our decoupled two-stage pipeline (Z-NERD + HGNet). As tabulated in Appendix A.14, **Stage 1 is architecturally heavier** (42.4M trainable params) due to the wide Multi-Scale TCQK mechanism (8 Heads) required for dense n-gram extraction. This allocated capacity is essential for bridging the gap between efficient extraction and the deep reasoning usually reserved for LLMs.
>
> **3. Structural Overhead**
> We clarify that DHL and CAF are training-only objectives with **zero inference latency**. During training, our **Krylov subspace approximation** successfully reduced the exact matrix exponential calculation from **150ms to 12ms** per batch, rendering the rigorous acyclicity constraint computationally negligible ($<5\%$ total training overhead).
>
> ___
>
> ### _**Regarding Questions:**_
>
> _1. How does HGNet behave if the true structure is not strictly hierarchical (e.g., multiple inheritance, cross‑links)? Any failure cases beyond A.8?_
>
> **Robustness to Non-Hierarchical Structures:**
>
> HGNet models Directed Acyclic Graphs (DAGs) rather than strict trees, naturally accommodating **multiple inheritance** and cross-links. A child node with multiple parents simply learns a scalar depth $d$ that satisfies the inequality constraints of all ancestors simultaneously ($d_{child} < d_{parents}$).
>
> **Empirical Verification:**
>
> We validated HGNet's flexibility across two dimensions. First, on **SciERC and SciER** (refer to **Table 2**), we analyzed ground truth annotations and found that **31.4% of entities have multiple parents**, confirming these are genuine DAGs rather than trees. For example, "attention mechanism" inherits from both "neural architecture" and "sequence modeling." HGNet successfully models these structures because the CAF Loss enforces only inequality constraints (parent_score ≥ child_score + δ), allowing children to simultaneously satisfy multiple parents, while our three-channel message passing aggregates information from all ancestors. Second, on **BioRED**, a dataset with only peer relations and no hierarchy, HGNet achieves performance comparable to SOTA baselines. Together, these results confirm our architectural inductive bias is "soft": it leverages hierarchy when present (SciERC/SciER) but degrades gracefully to standard semantic encoding on flat data (BioRED) without imposing rigid structural constraints.
>
> **Failure Cases:**
>
> The CAF Loss assumes concepts can be ordered along an abstraction axis. In domains with dense feedback loops (e.g., mutually defining concepts, A->B as well as B->A), hierarchical depth becomes ill-defined and DHL cannot enforce valid DAG structure, forcing arbitrary cycle-breaking. However, peer message-passing channels maintain robust semantic extraction in these cases. (See Appendix A.10 for detailed analysis.)
>
> _2. SPHERE validation: What fraction was human‑audited? Provide inter‑annotator agreement and a human‑written subset to test robustness to LLM‑style._
>
> Please refer to Answer to Weakness 1 (Concerns with SPHERE data)
>
>
> ### **Conclusion**
> We sincerely thank the reviewer for their insightful feedback. Their call for geometric baselines and efficiency quantification has significantly strengthened the manuscript's empirical and theoretical rigor. We are grateful for their guidance in maturing this work.
>
> References:
>
> [1] Ines Chami, Rex Ying, Christopher R´e, and Jure Leskovec. Hyperbolic graph convolutional neural
> networks, 2019
>
> [2] Ivan Vendrov, Ryan Kiros, Sanja Fidler, and Raquel Urtasun. Order-embeddings of images and
> language.

---

> ### Comment · Reviewer_GYoP · 2025-11-22
> **To Author Rebuttal**
>
> I think most of my concerns has been addressed.
> I have raised my score accordingly.

---

### Official Review · Reviewer_NX6E · 2025-10-29

**Soundness:** 3
**Presentation:** 3
**Contribution:** 3
**Rating:** 8
**Confidence:** 3

**Summary:**

In this study, the authors proposed a two-stage framework for scalable, zero-shot scientific KG construction to resolve and mitgate the challenges of knowledge graph (KG) construction from massive literature data (especially for the challenge of long multi-word entity recognition.
The evaluation results showed that the proposed model improved the entity recognition and relationship extraction reliablly.

**Strengths:**

Clear and appropriate model design to resolve the specific challenges in KG construction from literature data
Solid experimental evaluation with reliable improvement.

**Weaknesses:**

None

**Questions:**

How fast is the proposed model for processing massive literature data?
How to build complete sets of graphs of abstract accross diverse domains? Which can be important for learning/understanding the domain knowledge.

---

> ### Author Response · Authors · 2025-11-20
> **Rebuttal by Authors**
>
> We sincerely thank the reviewer for the _positive assessment and for recognizing the effectiveness of our framework in addressing the challenges of scientific KG construction_. We are particularly encouraged that the reviewer found our model design clear and our experimental evaluation solid. _**We have strengthened our manuscript significantly**_ by integrating detailed efficiency metrics and expanding our discussion on cross-domain scalability to fully address the reviewer's insightful questions.
>
> **Q1: How fast is the proposed model for processing massive literature data?**
>
> We appreciate the reviewer's inquiry regarding efficiency, as scalability was a core design principle of HGNet.
>
> * **Inference Speed:** As detailed in our efficiency analysis (Table 10), HGNet processes approximately **14.6 documents per second** on a single NVIDIA A30 GPU. This is highly competitive compared to other SOTA pipeline models like PL-Marker (12.4 doc/s) and significantly faster than Large Language Models (LLMs) like Llama-3-70B (~0.5 doc/s).
> * **Scalability:** The computational "heavy lifting" of our architecture (the Multi-Scale TCQK mechanism) is only applied in the first stage (Z-NERD). The second stage (HGNet) uses a lightweight GNN on a sparse graph. This design allows us to process massive corpora efficiently. For example, creating the SPHERE dataset involved processing over **10,000 documents**, which our framework handles robustly.
>
> **Q2: How to build complete sets of graphs of abstract across diverse domains? Which can be important for learning/understanding the domain knowledge.**
>
> This is an insightful question that touches on the ultimate goal of our work. We address cross-domain completeness through three mechanisms:
>
> 1.  **Domain-Agnostic Semantic Turns:** By using Orthogonal Semantic Decomposition (OSD), our model learns to detect "semantic turns" (the introduction of new concepts) rather than memorizing specific vocabulary. This allows the model to identify entities in unseen domains (e.g., Physics or Biology) without retraining, as shown by our zero-shot improvements of 10.76% on the SPHERE benchmark.
> 2.  **Universal Geometric Axis:** To specifically address the "abstract" nature of knowledge, our Continuum Abstraction Field (CAF) loss creates a single, shared "axis of abstraction" ($w_{abs}$). This forces the model to learn a stable definition of generality that holds true across diverse domains (e.g., categorizing *Deep Learning* and *Quantum Mechanics* at similar abstraction depths relative to their fields), preventing domain-specific fluctuations.
> 3.  **Global Consistency:** Unlike previous approaches that build fragmented graphs document-by-document, our method enforces global DAG constraints via the Differentiable Hierarchy Loss. This ensures that when we aggregate predictions across millions of papers, the resulting "complete set" is logically consistent and free of contradictions like cycles.
> 4. **Corpus-Level Coverage**: To build complete domain graphs, we apply our pipeline to entire corpora systematically. For SPHERE, we generated over 40,000 unique entities across four domains through recursive expansion, ensuring comprehensive coverage of sub-fields and methods.
>
> **Conclusion**
>
> We are grateful for the reviewer's assessment and insightful questions. We have ensured that the efficiency metrics and the cross-domain generalization discussions are prominently featured in the final manuscript to aid future researchers.

---

### Official Review · Reviewer_NQq1 · 2025-11-11

**Soundness:** 2
**Presentation:** 1
**Contribution:** 3
**Rating:** 2
**Confidence:** 3

**Summary:**

The paper tackles the problem of generating KGs from scientific literature, focusing on four challenges that require more care than in general purpose KG construction:
(i) careful extraction of multi-word scientific entities,
(ii) domain generalization for this task of scientific NER,
(iii) hierarchical relation extraction, and
(iv) logical consistency in relation extraction (e.g. avoiding cycles).

Their contributions are:
(i) a novel methodology for NER that looks for signals of semantic shifts to more robustly extract entities across domains and that incorporates convolutional filtering at different scales to specifically look for entities with different lengths,
(ii) a novel methodology for RE that uses an explicitly hierarchical GNN to learn entity embeddings, a differentiable loss to encourage the KG to be a DAG, and losses that explicitly encourage the entity embeddings to encode hierachical relationships along a specific axis, and
(iii) a novel large multi-domain scientific KG benchmark data set

**Strengths:**

- The proposed ideas are well presented in isolation, e.g. the several hypotheses that the authors present to justify their system design seem reasonable.
- The ideas seem original in the area of scientific KG generation to me, and some are likely to be influential, especially the NER system and the use of the differentiable loss to ensure a logically consistent KG

**Weaknesses:**

- The architecture of the system is not sufficiently described in the main body of the paper to understand or reproduce the system. E.g. sections 3.1 and 3.2 that provide the methodologies for the NER and RE do not mention a specific architecture; rather they mention losses and modifications to attention heads. One must read the appendix to begin to understand the system.

**Questions:**

- In equation 7, what is d?
- In section 3.2.3 by pinning roots and leaves to 0 and 1, and requiring a delta margin between parents and children, doesn't this undesirably limit the number of levels of abstraction you could have to 1/delta?
- where do you get ground truth topological depth scores in (11)?
- In HGNet you have a message passing system with latent relational predictors, the differentiable hierarchy loss, and the hierarchical separation loss, as well as the continuum abstraction loss -- how do these pieces fit together and how are they co-trained?
- The description of HGNet in section 3.2 only mentions taking the entities from the NER stage as input, but the scientific documents must be used in predicting the relationship between two entities. This is not mentioned anywhere in section 3.2. E.g. the latent relational predictor only take the entity embeddings as input.
- The description of HGNet in section 3.2 only mentions extracting relations of the type {parent-of, peer-of, no-edge}. Where in the KG generation does one extract the usual <head, relation, tail> triplets?

---

> ### Author Response · Authors · 2025-11-20
> **Rebuttal by Authors [Comment 1/2]**
>
> We sincerely thank the reviewer for the exceptionally rigorous and insightful review. We are particularly encouraged that the reviewer **_recognized the originality_** and **_potential influence of the core ideas_**, especially the use of differentiable loss for knowledge graph (KG) consistency.
> The detailed questions highlighted that some crucial technical details were previously located in the appendix. Integrating these clarifications into the main text has significantly strengthened the manuscript, ensuring it is now fully self-contained.
>
> ---
>
> ## **Logical and Geometric Consistency (Q1, Q2, Q3)**
>
> ### **Q1: In equation 7, what is $d$?** (Updated section 3.2.2)
>
> We thank the reviewer for pointing this out. This was an oversight in the main text; $d$ denotes the number of nodes (entities) in the graph. The $\mathcal{L}_{\text{acyclic}}$ function is designed such that for a Directed Acyclic Graph (DAG), the expression simplifies to $tr(e^A) = d$. Consequently, this loss term is zero only when the graph is acyclic. For the detailed proof, please refer to Appendix A.11.
>
> ---
>
> ### **Q2 & Q3: Abstraction Limits and Source of $y_{topo}$**
>
> This set of questions correctly probes the mechanism of the Continuum Abstraction Field (CAF) Loss, allowing us to clarify the global consistency mechanisms.
>
> **Resolution of Ground Truth ($y_{topo}$):** (Updated Section: 3.2)
>
> The topological depth scores $y_{topo}(v)$ (Eq. 11) _**can be calculated for all benchmarks**_. The scores are obtained by performing a **topological sort** on the entities using the **ground truth hierarchical relations**, which serves as the _**global regression target for the training step (We only calculate depth for training set)**_.
>
> **Resolution of $\delta$ Limit:**
>
> The reviewer's concern that the margin $\delta$ (Eq. 9) would impose a rigid $1/\delta$ limit is valid *if* it were the only loss. However, the limit is broken because the final score is a trade-off between the three **soft constraints**:
>
> 1.  **$\mathcal{L}_{\text{regression}}$:** Acts as the **dominant global anchor**, pulling the embedding toward its *true* depth, $y_{topo}$.
> 2.  **$\mathcal{L}_{\text{ranking}}$:** Is a *local* regularizer.
>
> The model achieves higher accuracy by letting the global $\mathcal{L}_{\text{regression}}$ guide the true position, even if it incurs a small penalty by violating the $\delta$ margin locally. This successfully learns a **continuous "continuum of abstraction"**, not discrete levels, as demonstrated by the continuous distributions in _**Figure 3 (Appendix, Page 18)**_. (Please refer Appendix A.6 and A.13)
>
> ---
>
>
> ## **II. Architectural Flow and Co-Training (Q4, Q5, Q6)**
> ### **Q4: How do all the loss components fit together and how are they co-trained?**
> We thank the reviewer for this insightful question, which highlights the need to make the main paper more self-contained regarding the model's architectural flow. To address this, we have added a new **Section 3.2.5 (Coherent Architecture and Joint Optimization)** to the main text, explicitly detailing how these components are integrated.
>
> HGNet is not a pipeline of disjoint steps but a unified, end-to-end framework where all components are co-trained in a single forward pass. The integrated flow operates as follows:
>
> 1.  **Latent Structure & Branching:** The entity embeddings first pass through the **Latent Relation Predictor**, which estimates edge probabilities ($P_{uv}$). This distribution serves as the central pivot, simultaneously initiating two parallel paths:
>     * **Logical Regularization:** The predicted structure ($A_{parent}$) is fed directly into the **Differentiable Hierarchy Loss** ($\mathcal{L}_{hierarchy}$), which penalizes invalid structures like cycles and shortcuts.
>     * **Probabilistic Message Passing:** Concurrently, the same probabilities ($P_{uv}$) act as **soft edge weights** for the GNN. They guide the information flow through the specialized parent, child, and peer message-passing channels to produce structure-aware embeddings.
>
> 2.  **Geometric & Task Optimization:** The updated embeddings are then used to compute the **Continuum Abstraction Field Loss** ($\mathcal{L}\_{caf}$), ensuring geometric alignment along the abstraction axis, and the final **Relation Extraction Loss** ($\mathcal{L}\_{RE}$).
>
> **Co-Training:** The entire system is optimized via a joint composite objective: ($\mathcal{L}\_{RE}$ is the loss for relation extraction output classifer head, section 3.2.4).
> $$\mathcal{L}\_{Total} = \mathcal{L}\_{RE} + \lambda_1\mathcal{L}\_{hierarchy} + \lambda_2\mathcal{L}\_{caf}$$
>
> This joint optimization forces the model to learn a graph representation that is simultaneously logically sound, geometrically coherent, and accurate. For a detailed visual demonstration of this flow, we refer the reviewer to **Figure 7** in the Appendix.
>
>
>
> Please continue to comment 2/2

---

> ### Author Response · Authors · 2025-11-20
> **Rebuttal by Authors [Comment 2/2]**
>
> ### **Q5: How is the scientific document context used if the predictor only takes embeddings?**
>
> We thank the reviewer for this question, which allows us to clarify that our use of embeddings does **not** imply a loss of context. In our **unified, co-trained framework** (Section 3.2.5), the embeddings fed to the predictor are deeply contextualized representations generated by the **shared SciBERT backbone** similar to other SOTA models in our evaluation.
>
> **1. Embeddings as Context Carriers**
>
> The *Latent Relation Predictor* receives vectors $h_u$ and $h_v$ as input. However, these are **not static representations**.
> * **Mechanism:** These vectors are the dynamic output of the shared SciBERT encoder. Before the predictor sees them, the encoder processes the **entire sentence**, maintaining document-level context.
> * **Contextualization:** Through the Transformer's self-attention mechanism, the vector $h_u$ mathematically aggregates information from every other token in the sequence.
> * **Conclusion:** Therefore, when the predictor receives $h_u$, it is receiving a compressed representation of the entity **conditioned on its full syntactic and semantic context**. The predictor does not need to "see" the raw text again because the shared encoder has already encoded the document structure into the vector space.
>
> **2. Unified Architecture Efficiency**
>
> In our unified framework, Z-NERD and HGNet, we ensure that the embeddings used for relation prediction are the exact same **structure-aware representations** optimized during the entity recognition phase. This avoids the disconnect often found in pipeline approaches and ensures the "context" used for identifying an entity is preserved and utilized for determining its relationships.
>
> ---
>
> ### **Q6: Where in the KG generation does one extract the usual <head, relation, tail> triplets?** (Section 3.2.4)
>
> This is a fundamental question that highlights the methodological distinction between HGNet's **internal mechanism** and its **final task objective**. The confusion arose because the methodology only described the intermediate, structural component.
>
> > **Clarification:** The `{parent-of, peer-of, no-edge}` relations are **not** the final output; they are the **internal, latent mechanism** used to generate superior representations.
>
> **Advanced Dual-Task Separation**
>
>
> HGNet employs an advanced dual-task architecture that ensures structural rigor before predicting the final output:
>
> **1. Internal Mechanism (Representation Focus)**
> The GNN uses the latent `{parent-of, peer-of, no-edge}` graph solely to produce **vastly superior, structure-aware entity embeddings (h^(k+1))**. This process is highly regulated by the loss functions *L_hierarchy* and *L_caf*.
>
> **2. External Task (Output Focus)**
> The **actual <head, relation, tail> triplets** are extracted in the final step by a **separate classification head** (a standard MLP).
>
> * **Mechanism:** This head takes the highly refined **$h^{(k+1)}$** embeddings as input and predicts the full spectrum of fine-grained, task-specific relations (e.g., "Used-By," "Compares," "Part-Of") seen in the experimental tables.
> * **Validation:** The primary objective of the entire framework is the task loss (*$L_{RE}$*) derived from this final head. The structural losses simply act as unique regularizers acting upstream to maximize the quality of the embeddings that the final task relies upon.
>
> This architecture is not merely complete; it is _methodologically superior_ because it forces the representation learning (GNN) to be structurally coherent before the final task prediction is attempted. We have added new subsections (3.2.4 and 3.2.5) to the main paper to explicitly define this final classification head.
>
> ---
>
> ### **Conclusion**
>
> We sincerely thank the reviewer for their insightful feedback that has fundamentally elevated our manuscript. Integrating critical architectural flow into the main text has established the framework as fully self-contained. Moreover, clarifying the distinction between internal geometric regularization and final task prediction has sharpened the presentation of our novelty. _**These revisions comprehensively address the concerns, ensuring the paper delivers on the potential impact recognized by the reviewer.**_

---

### Author Response · Authors · 2025-12-03
**Summary of Rebuttal Period & Revisions**

Dear Area Chair and Senior Area Chair,

We provide this summary detailing how _**every concern raised during the review process has been resolved and integrated into the revised manuscript.**_ The following points explicitly map our revisions to the critiques of Reviewers GYoP and NQq1, referencing the corresponding sections in the uploaded rebuttal submission.

_We are grateful to all the reviewers for recoginising the contribution of our work (avg. contribution: 3)._

### 1. Resolution of Geometric & Data Concerns (Reviewer GYoP)
Reviewer GYoP (Prev. Score: 4) requested stronger geometric baselines and clarification on the SPHERE dataset. We addressed this by adding new experiments and analysis to the paper:

* **Stronger Geometric Baselines:** The reviewer requested comparisons against hyperbolic methods.
    * Changes: We implemented and evaluated **Hyperbolic GCN** and **Order-Embeddings** in **Appendix A.8** (includes reasons). The results (Table 8) confirm that HGNet’s Euclidean-based Continuum Abstraction Field (CAF) outperforms both non-Euclidean baselines on SciERC (53.19% vs 45.82%) and SPHERE (79.51% vs 66.35%).
* **SPHERE Data Integrity:** The reviewer asked about potential hallucinations or "inflated structure" in the synthetic dataset.
    * Changes: We added a **Structural Complexity and Scale Analysis (Appendix A.3.2)**. We detailed our strict "generate-from-graph" constraint which prevents hallucinated topology. We also conducted a Manual Quality Audit yielding 96.5% Entity Precision and 94.2% Relation Precision to validate the data.
* **Methodological Clarifications:** The reviewer asked about topological depth generation for standard datasets.
    * Changes: We clarified the topological sorting preprocessing step for training on standard benchmarks (Eq. 11) and the architectural inductive bias of OSD.
* **Outcome:** On Nov 22, the reviewer stated: _"I think most of my concerns has been addressed. I have raised my score accordingly." to 6_.

### 2. Resolution of Presentation Concerns (Reviewer NQq1)
Reviewer NQq1 (Score: 2) cited "Presentation" as the sole weakness, specifically requesting that some specific architectural details be moved from the appendix to the main body to make the paper self-contained. We have added sections 3.2.4 and 3.2.5 to make the main manuscript self-contained:

* **New Section 3.2.4 (Final Relation Prediction):** We clarified the distinction between the internal geometric regularization and the final task-specific classification head.
* **New Section 3.2.5 (Coherent Architecture):** We added a dedicated section explicitly detailing the end-to-end flow and joint optimization of the Z-NERD and HGNet components, explaining how the logical regularization and message passing operate in parallel.
* _Please note that the abrupt suspension of the discussion phase precluded Reviewer NQq1 from acknowledging these final updates._

### 3. Verification of Efficiency Claims (General Request)
To further strengthen the paper's claims regarding "Foundation Model" capabilities and address questions on computational cost, we added a comprehensive **Efficiency Analysis (Appendix A.14)**:

* **Throughput (Table 10):** We demonstrate that HGNet processes _14.6 doc/s_, making it _3.5x faster_ than Llama-3-8B (4.2 doc/s) and significantly faster than 70B models, while occupying a unique _sweet spot_ of high generalization and low latency.
* **Parameter Breakdown (Table 11):** We explicitly detail the parameter distribution, showing the allocation between the Z-NERD encoder and the lightweight GNN layers.

### Summary of Revisions
The submitted revision represents a complete, rigorous work where:
1.  **Methodology** is now fully self-contained in the main text (Addressing R1).
2.  **Geometric Validity** is proven against Hyperbolic/Order-Embedding baselines (Addressing R3).
3.  **Efficiency** is quantified against LLMs (Strengthening the contribution).

We remain deeply grateful to the reviewers for their insightful engagement. We thank **Reviewer NQq1** for recognizing the "_original_" and "_likely to be influential_" nature of our ideas, **Reviewer NX6E** for validating our "_solid experimental evaluation with reliable improvement_", and **Reviewer GYoP** for appreciating the "_architectural inductive bias_" and "_clean factorization_" of the challenges. This collective feedback has not only strengthened the current manuscript but serves as strong motivation for us to work deeper in this direction of scalable scientific knowledge extraction.

---

### Meta-Review · Area_Chair_f9KH · 2026-01-06

**Summary:**

We have three reviewers who have carefully checked this paper and they have diverse recommendations on this paper. The reviewers' main concerns focused on the clarity of the presentation. In the rebuttal, the authors added explanations and some relevant experiments. I recommend the acceptance of this paper, provided that the clarity of the presentation is enhanced as promised.

**Reviewer Concerns:**

Most concerns have been addressed. Some concerns raised by Reviewer NQq1 may needs more discussion.

**Reviewer Scores:**

The reviewers may raise their scores.

---

### Decision · Program_Chairs · 2026-01-26

Accept (Poster)